# Protein Sequence and Structure Co-Design with Equivariant Translation

**Chence Shi**[1,2,3], **Chuanrui Wang**[2,3], **Jiarui Lu**[2,3], **Bozitao Zhong**[2,3], **Jian Tang**[1,2,4,5]

[1]BioGeometry [2]Mila - Québec AI Institute [3]Université de Montréal
[4]HEC Montréal [5]CIFAR AI Research Chair

`chence.shi@{biogeom.com,umontreal.ca}`
`{chuanrui.wang,jiarui.lu,bozitao.zhong}@umontreal.ca`
`jian.tang@hec.ca`

## Abstract

Proteins are macromolecules that perform essential functions in all living organisms. Designing novel proteins with specific structures and desired functions has been a long-standing challenge in the field of bioengineering. Existing approaches generate both protein sequence and structure using either autoregressive models or diffusion models, both of which suffer from high inference costs. In this paper, we propose a new approach capable of protein sequence and structure co-design, which iteratively translates both protein sequence and structure into the desired state from random initialization, based on context features given *a priori*. Our model consists of a trigonometry-aware encoder that reasons geometrical constraints and interactions from context features, and a roto-translation equivariant decoder that translates protein sequence and structure interdependently. Notably, all protein amino acids are updated in one shot in each translation step, which significantly accelerates the inference process. Experimental results across multiple tasks show that our model outperforms previous state-of-the-art baselines by a large margin, and is able to design proteins of high fidelity as regards both sequence and structure, with running time orders of magnitude less than sampling-based methods.

## 1 Introduction

Proteins are macromolecules that mediate the fundamental processes of all living organisms. For decades, people are seeking to design novel proteins with desired properties (Huang et al., 2016), a problem known as *de novo protein design*. Nevertheless, the problem is very challenging due to the tremendous search space of both sequence and structure, and the most well-established approaches still rely on hand-crafted energy functions and heuristic sampling algorithms (Leaver-Fay et al., 2013; Alford et al., 2017), which are prone to arriving at suboptimal solutions and are computationally intensive and time-consuming.

Recently, machine learning approaches have demonstrated impressive performance on different aspects of protein design, and significant progress has been made (Gao et al., 2020). Most approaches use deep generative models to design protein sequences based on corresponding structures (Ingraham et al., 2019; Jing et al., 2021; Hsu et al., 2022). Despite their great potential for protein design, the structures of proteins to be engineered are often unknown (Fischman & Ofran, 2018), which hinders the application of these methods. Therefore, efforts have been made to develop models that *co-design* the sequence and structure of proteins (Anishchenko et al., 2021; Wang et al., 2021). As a pioneering work, Jin et al. (2021) propose an autoregressive model that co-designs the Complementarity Determining Regions (CDRs) sequence and structure of antibodies based on iterative refinement of protein structures, which spurs a lot of follow-up works (Luo et al., 2022; Kong et al., 2022). Nevertheless, these approaches are tailored for antibodies and their effectiveness remains unclear on proteins with arbitrary domain topologies (Anand & Achim, 2022). In addition, they often suffer from high inference costs due to autoregressive sampling or annealed diffusion sampling (Song & Ermon, 2019; Luo et al., 2022). Very recently, Anand & Achim (2022) propose another diffusion-based generative model (Ho et al., 2020) for general protein sequence-structure co-design, where they

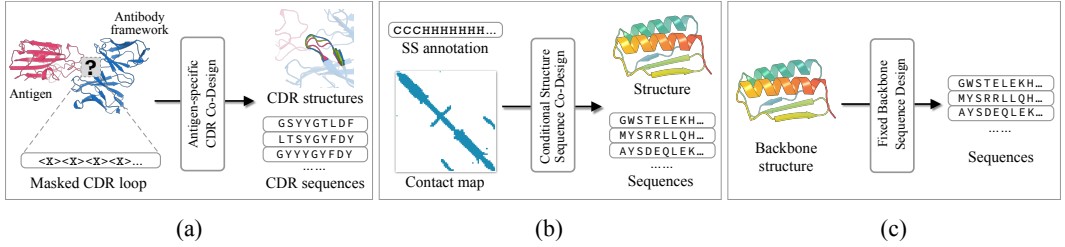

Figure 1: Illustration of three protein design tasks with different context features. (a) Antigen-specific CDR co-design given structure and sequence of antibody framework and the binding antigen. (b) Protein sequence-structure co-design conditioned on secondary structure (SS) annotation and binary contact features. (c) Fixed backbone sequence design conditioned on given backbone structures.

adopt three diffusion models to generate structures, sequences, and rotamers of proteins in sequential order. Although applicable to proteins of all topologies, such a sequential generation strategy fails to cross-condition on sequence and structure, which might lead to inconsistent proteins. Besides, the inference process is also expensive due to the use of three separate diffusion processes.

To address the aforementioned issues, in this paper, we propose a new method capable of protein sequence-structure equivariant co-design called PROTSEED. Specifically, we formulate the co-design task as a translation problem in the joint sequence-structure space based on context features. Here the context features represent prior knowledge encoding constraints that biologists want to impose on the protein to be designed (Dou et al., 2018; Shen et al., 2018). As an illustration, we present three protein design tasks with different given context features in Figure 1. Our PROTSEED consists of a trigonometry-aware encoder that infers geometrical constraints and prior knowledge for protein design from context features, and a novel roto-translation equivariant decoder that iteratively translates proteins into desired states in an end-to-end and equivariant manner. The equivariance property with respect to protein structures during the whole process is guaranteed by predicting structure updates in local frames based on invariant representations, and then transforming them into global frames using change of basis operation. It is worth mentioning that PROTSEED updates sequence and structure of all residues in an one-shot manner, leading to a much more efficient inference process. In contrast to previous method that first generates structure and then generates sequence and rotamers, we allow the model to cross-condition on sequence and structure, and encourage the maximal information flow among context features, sequences, and structures, which ensure the fidelity of generated proteins.

We conduct extensive experiments on the Structural Antibody Database (SAbDab) (Dunbar et al., 2014) as well as two protein design benchmark data sets curated from CATH (Orengo et al., 1997), and compare PROTSEED against previous state-of-the-art methods on multiple tasks, ranging from antigen-specific antibody CDR design to context-conditioned protein design and fixed backbone protein design. Numerical results show that our method significantly outperforms previous baselines and can generate high fidelity proteins in terms of both sequence and structure, while running orders of magnitude faster than sampling-based methods. As a proof of concept, we also show by cases that PROTSEED is able to perform *de novo* protein design with new folds.

## 2 RELATED WORK

**Protein Design.** The most well-established approaches on protein design mainly rely on handcrafted energy functions to iteratively search low-energy protein sequences and conformations with heuristic sampling algorithms (Leaver-Fay et al., 2013; Alford et al., 2017; Tischer et al., 2020). Nevertheless, these conventional methods are computationally intensive, and are prone to arriving at local optimum due to the complicated energy landscape. Recent advances in deep generative models open the door to data-driven approaches, and a variety of models have been proposed to generate protein sequences (Rives et al., 2021; Shin et al., 2021; Ferruz et al., 2022) or backbone structures (Anand & Huang, 2018; Eguchi et al., 2022; Trippe et al., 2022). To have fine-grain control over designed proteins, methods are developed to predict sequences that can fold into given backbone structures (Ingraham et al., 2019; Jing et al., 2021; Anand et al., 2022; Dauparas et al., 2022), a.k.a. fixed backbone design, which achieve promising results but require the desired protein structure to be known *a priori*.

Recently, a class of approaches that generate both protein sequence and structure by network hallucination have emerged (Anishchenko et al., 2021; Wang et al., 2021), which carry out thousands of gradient descent steps in sequence space to optimize loss functions calculated by pre-trained protein structure prediction models (Yang et al., 2020; Jumper et al., 2021). However, the quality of designed proteins usually relies on the accuracy of structure prediction models, and is sensitive to different random startings. On the other hand, attempts have been made to co-design CDR sequence and structure of antibodies using either autoregressive models (Saka et al., 2021; Jin et al., 2021) or diffusion models (Luo et al., 2022). Nevertheless, they are restricted to proteins with specific domain topologies and often suffer from the time-consuming Monte Carlo sampling process. Going beyond antibodies, Anand & Achim (2022) adopt three separate diffusion models to generate sequences, structures, and rotamers of proteins sequentially. Such a method is inefficient and fails to cross-condition on both protein sequence and structure. Our model also seeks to co-design protein sequence and structure, but is able to cross-condition on sequence and structure, while being much more efficient.

**3D Structure Prediction.** Our work is also related to approaches that perform 3D structure prediction by iteratively translating structures in three dimensional space equivariantly (Shi et al., 2021; Luo & Hu, 2021; Hoogeboom et al., 2022; Xu et al., 2022; Zhu et al., 2022). However, these methods represent structures as either molecular graphs or point clouds, and are not applicable to protein structures. On the other hand, protein folding models (Jumper et al., 2021; Baek et al., 2021) that perform protein structure prediction require complete protein sequences as well as their Multiple Sequence Alignments (MSAs) as input, and cannot co-design protein sequence and structure directly.

## 3    METHOD

### 3.1    PRELIMINARIES

**Notations.** Proteins are macromolecules that can be viewed as chains of amino acids (residues) connected by peptide bonds. In this paper, an amino acid can be represented by its type $s_i \in \{1, \cdots 20\}$, $C_\alpha$ coordinates $\boldsymbol{x}_i \in \mathbb{R}^3$, and the frame orientation $\boldsymbol{O}_i \in \text{SO}(3)$, where $i \in \{1, \cdots N\}$ and $N$ is the number of residues in a protein. The $\boldsymbol{x}_i$ and $\boldsymbol{O}_i$ form a canonical orientation frame with respect to the N, C and $C_\beta$ atoms, from which the backbone atom positions can be derived. We denote the one-hot encoding of the residue type as $\boldsymbol{s}_i = \text{onehot}(s_i)$. In the protein sequence and structure co-design task, researchers often provide context features as input to encourage designed proteins to have desired structural properties. These context features can either be single (residue) features $\boldsymbol{m}_i \in \mathbb{R}^{c_m}$ (e.g., amino acid secondary structure annotations) or pair features $\boldsymbol{z}_{ij} \in \mathbb{R}^{c_z}$ (e.g., binary contact features between residues). With the above notations, a protein with $N$ residues can be compactly denoted as $\mathcal{P} = \{(\boldsymbol{s}_i, \boldsymbol{x}_i, \boldsymbol{O}_i)\}_{i=1}^N$. The context features known *a priori* can be denoted as $\{\boldsymbol{m}_i\} \in \mathbb{R}^{N \times c_m}$ and $\{\boldsymbol{z}_{ij}\} \in \mathbb{R}^{N \times N \times c_z}$.

**Problem Formulation.** Given a set of context features $\{\boldsymbol{m}_i\} \in \mathbb{R}^{N \times c_m}$ and $\{\boldsymbol{z}_{ij}\} \in \mathbb{R}^{N \times N \times c_z}$, the task of *protein sequence and structure co-design* is the joint generation of residue types and 3D conformations of a protein with $N$ residues, i.e., the conditional generation of $\mathcal{P} = \{(\boldsymbol{s}_i, \boldsymbol{x}_i, \boldsymbol{O}_i)\}_{i=1}^N$ based on $\{\boldsymbol{m}_i\}$ and $\{\boldsymbol{z}_{ij}\}$. Note that context features vary from setting to setting. For example, in antibody CDR design (Jin et al., 2021; Luo et al., 2022), they are derived from antibody framework and binding antigen structures with CDR region masked, while in full protein design (Anand & Achim, 2022), they can be secondary structure annotations and residue-residue contact features.

**Overview.** In this paper, we formulate protein sequence and structure co-design as an equivariant translation problem in the joint sequence-structure space. Specifically, we develop a trigonometry-aware context encoder to first reason geometrical constraints encoded in context features. Based on the updated context features, protein sequence and structure are jointly generated in an iterative manner by a novel roto-translation equivariant decoder, starting from randomly initialized structures and residue types (illustrated in Figure 2). To model interactions between sequence and structure during decoding, we allow information to flow between context features, structures, and residue types in each translation step. In this way, the generated sequence and structure are ensured to be consistent. The pseudo-code of the whole framework can be found in Algorithm 1. The rest of this section is organized as follows: Section 3.2 introduces the trigonometry-aware context encoder. Section 3.3 elaborates the iterative joint sequence-structure decoder and training objectives.

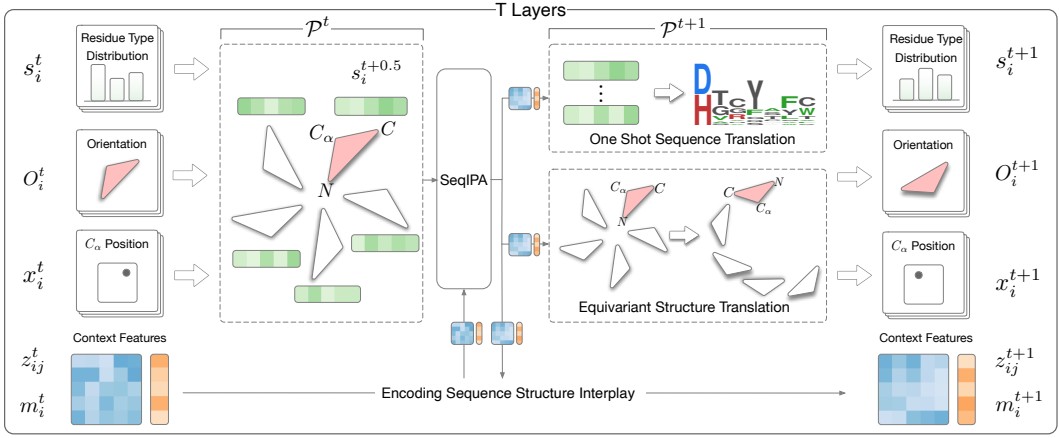

Figure 2: Illustration of the joint sequence-structure translation process. In each translation layer, the network first captures the interactions of the current protein state and context features via SeqIPA, and then translates the protein sequence and structure into the next state equivariantly.

## 3.2 TRIGONOMETRY-AWARE CONTEXT ENCODER

Given single features and pair features as input, the goal of the context encoder is to capture the interactions between different context features and infer encoded constraints for the following protein sequence-structure co-design. We first embed single features and pair features into $c$ dimensional space using Multiple Layer Perceptrons (MLPs). We then adopt a stack of $L$ trigonometry-aware update layer to propagate information between single features and pair features.

We represent the updated single features and pair features at $l^{\text{th}}$ layer as $\{\boldsymbol{m}_i^l\}$ and $\{\boldsymbol{z}_{ij}^l\}$, respectively. At each layer, the single features are first updated using a variant of multi-head self-attention (denoted as MHA) (Vaswani et al., 2017), with pair features serving as additional input to bias the attention map. Similar to Jumper et al. (2021), the pair feature $\boldsymbol{z}_{ij}^l$ are then updated by the linear projection of outer product of single features $\boldsymbol{m}_i^{l+1}$ and $\boldsymbol{m}_j^{l+1}$:

$$\boldsymbol{m}_i^{l+1} = \text{MHA}(\{\boldsymbol{m}_i^l\}, \{\boldsymbol{z}_{ij}^l\}), \tag{1}$$

$$\boldsymbol{z}_{ij}^{l+0.5} = \boldsymbol{z}_{ij}^l + \text{Linear}(\boldsymbol{m}_i^{l+1} \otimes \boldsymbol{m}_j^{l+1}), \tag{2}$$

where $\otimes$ is the outer product operation. Notably, we enable the information to flow between single features and pair features to better model interactions between context features.

Since pair features (e.g., contact map and distance map) are usually related with Euclidean distances and dihedral angles between residues, inspired by AlphaFold 2 (Jumper et al., 2021), we adopt two trigonometry-aware operations (Eq. 4 and Eq. 5) in each layer to maintain geometric consistency and encourage pair features to satisfy the triangle inequality. Formally, we have:

$$\hat{\boldsymbol{a}}_{ij}, \hat{\boldsymbol{b}}_{ij} = \sigma(\boldsymbol{z}_{ij}^{l+0.5}) \odot \text{Linear}(\boldsymbol{z}_{ij}^{l+0.5}), \quad \boldsymbol{q}_{ij}, \boldsymbol{k}_{ij}, \boldsymbol{v}_{ij}, \boldsymbol{b}_{ij} = \text{Linear}(\boldsymbol{z}_{ij}^{l+0.75}), \tag{3}$$

$$\boldsymbol{z}_{ij}^{l+0.75} = \boldsymbol{z}_{ij}^{l+0.5} + \sigma(\boldsymbol{z}_{ij}^{l+0.5}) \odot \text{Linear}\Big(\sum_k \hat{\boldsymbol{a}}_{ik} \odot \hat{\boldsymbol{b}}_{jk} + \hat{\boldsymbol{a}}_{ki} \odot \hat{\boldsymbol{b}}_{kj}\Big), \tag{4}$$

$$\boldsymbol{z}_{ij}^{l+1} = \boldsymbol{z}_{ij}^{l+0.75} + \sigma(\boldsymbol{z}_{ij}^{l+0.75}) \odot \sum_k (\alpha_{ijk}\boldsymbol{v}_{ik} + \alpha_{ijk}\boldsymbol{v}_{kj}), \tag{5}$$

where $\sigma(\cdot) = \text{sigmoid}(\text{Linear}(\cdot))$, and $\alpha_{ijk} = \text{softmax}_k\big(\frac{1}{\sqrt{c}}\boldsymbol{q}_{ij}^\top(\boldsymbol{k}_{ik} + \boldsymbol{k}_{kj}) + \boldsymbol{b}_{jk} + \boldsymbol{b}_{ki}\big)$ is the attention score of a novel trigonometry-aware attention. Intuitively, in our trigonometry-aware attention, the pair feature $\boldsymbol{z}_{ij}$ is updated with neighboring features $\boldsymbol{z}_{ik}$ and $\boldsymbol{z}_{kj}$, by enumerating all possible $k$ that form a triangle $\Delta_{ijk}$ in terms of residues. Different from Jumper et al. (2021), we tie the attention score $\alpha_{ijk}$ within each triangle $\Delta_{ijk}$ to reduce the computational burden, while keeping the whole network sensitive to triangular interactions among residues. After $L$ rounds of feature propagation, the updated single features $\{\boldsymbol{m}_i^L\}$ and pair features $\{\boldsymbol{z}_{ij}^L\}$ serve as inputs to the decoder for joint protein sequence-structure design.

### 3.3 JOINT SEQUENCE-STRUCTURE DECODER

In this section, we describe the proposed joint sequence-structure decoder, with the goal of iteratively translating protein sequence and structure into desired states from scratch based on context features. Simply parameterizing the decoder using two neural networks that generate sequence and structure separately is problematic, as we need to ensure the consistency between generated sequence and structure, i.e., the sequence folds to the structure. Meanwhile, we require our decoder to be roto-translation equivariant (Köhler et al., 2020; Jing et al., 2021) with respect to all protein structures during the decoding. To this end, we develop a novel roto-translation equivariant network composed of $T$ consecutive translation layers with weight tying (Dehghani et al., 2018; Bai et al., 2019). In each layer, we update context features, residue structures, and residue types interdependently, by allowing information to propagate among them. It is worth mentioning that residue types and residue structures of all amino acids are updated in one shot in each translation step, distinct from previous work that generates them autoregressively, which significantly accelerates the decoding procedure.

We represent the updated residue types and residue structures at $t^{\text{th}}$ layer as $\mathcal{P}^t = \{(s_i^t, x_i^t, O_i^t)\}_{i=1}^N$. Abusing the notation a little bit, we denote updated context features at $t^{\text{th}}$ layer as $\{m_i^t\}$ and $\{z_{ij}^t\}$. Specifically, $\{m_i^0\}$ and $\{z_{ij}^0\}$ are the invariant output of context encoder introduced in Section 3.2. For each amino acid, we initialize the residue type as the uniform distribution over 20 amino acid types, i.e., $s_i^0 = \frac{1}{20} \cdot \mathbf{1}$, $C_\alpha$ coordinates as origin in global frame, i.e., $x_i^0 = (0, 0, 0)$, and frame orientation as identity rotation, i.e., $O_i^0 = I_3$. We elaborate each translation layer next.

**Encoding Sequence-Structure Interplay.** In each translation layer, we start by embedding current residue types into $c$ dimensional space with a feedforward network $\text{MLP}_{\text{e}} : \mathbb{R}^{20} \to \mathbb{R}^c$, $s_i^{t+0.5} = \text{MLP}_{\text{e}}(s_i^t)$. We then adopt a variant of Invariant Point Attention (IPA) (Jumper et al., 2021) called SeqIPA to capture the interplay of residue types, residue structures and context features, integrating them all altogether into updated context features. Such a practice is often favored in literature (Anand & Achim, 2022; Luo et al., 2022; Tubiana et al., 2022) as it is aware of the orientation of each residue frame while being roto-translation invariant to input and output features. Distinct from vanilla IPA, our SeqIPA takes residue types as the additional input to bias the attention map and steer the representation of the whole protein generated so far:

$$m_i^{t+1}, z_{ij}^{t+1} = \text{SeqIPA}(\{m_i^t\}, \{z_{ij}^t\}, \{s_i^{t+0.5}\}, \{x_i^t\}, \{O_i^t\}). \tag{6}$$

Note that SeqIPA is orientation-aware with respect to residue structures and roto-translation invariant to all other representations. We refer readers to Appendix B.2 for more details about the SeqIPA.

**Equivariant Structure Translation.** Given updated context features, the protein structure is translated towards the next state by updating $\{x_i^t\}$ and $\{O_i^t\}$. To update $C_\alpha$ positions $\{x_i^t\}$, we first predict the change of coordinates (denoted as $\{\hat{x}_i^t\}$) within each local residue frame specified by $\{O_i^t\}$. We then perform a change of basis using $\{O_i^t\}$ to transform $\{\hat{x}_i^t\}$ from local frame into global frame (Kofinas et al., 2021; Hsu et al., 2022) to derive equivariant deviation of $C_\alpha$ positions (Eq. 7). Intuitively, the deviation of $C_\alpha$ positions rotate accordingly when residue frames rotate, which guarantees the equivariance of the $C_\alpha$ translation step.

The update for orientation frame $O_i^t$ is computed by predicting a unit quaternion vector (Jia, 2008) with a feedforward network, which is then converted to a rotation matrix $\hat{O}_i^t$, and left-multiplied by $O_i^t$ to rotate the residue frame (Eq. 8). We adopt the unit quaternion here because it is a more concise representation of a rotation in 3D than a rotation matrix. Since the predicted unit quaternion is an invariant vector, the predicted rotation matrix $\hat{O}_i^t$ is also invariant. Therefore, the translation step of the residue frame is equivariant due to the multiplication of rotation matrices. We summarize the whole equivariant structure translation step as follows:

$$\hat{x}_i^t = \text{MLP}_{\text{x}}(m_i^{t+1}, m_i^0), \quad x_i^{t+1} = x_i^t + O_i^t \hat{x}_i^t, \tag{7}$$

$$\hat{O}_i^t = \text{convert}\left(\text{MLP}_{\text{o}}(m_i^{t+1}, m_i^0)\right), \quad O_i^{t+1} = O_i^t \hat{O}_i^t, \tag{8}$$

where convert is a function that converts a quaternion to a rotation matrix.

**One Shot Sequence Translation.** The residue type of all amino acids are updated in one shot based on updated context features and current residue types in each translation step. Specifically, we use a feedforward network $\text{MLP}_{\text{s}}$ to predict residue type distributions over 20 amino acid types for the next iteration:

$$s_i^{t+1} = \text{softmax}\left(\lambda \cdot \text{MLP}_{\text{s}}(m_i^{t+1}, m_i^0, s_i^{t+0.5})\right), \tag{9}$$

where $\lambda$ is a hyper-parameter controlling the temperature of the distribution. See Appendix B.3 for discussions on hyper-parameters and the equivariance property of sequence and structure translation.

Summarizing the above, at $(t + 1)^{\text{th}}$ layer, the decoder takes $\mathcal{P}^t = \{(\boldsymbol{s}_i^t, \boldsymbol{x}_i^t, \boldsymbol{O}_i^t)\}_{i=1}^N$ as the input and computes $\mathcal{P}^{t+1} = \{(\boldsymbol{s}_i^{t+1}, \boldsymbol{x}_i^{t+1}, \boldsymbol{O}_i^{t+1})\}_{i=1}^N$ as the output. Based on $\mathcal{P}^t$, we can efficiently reconstruct full backbone atom positions according to their averaged relative positions with respect to $C_\alpha$ recorded in literature (Engh & Huber, 2012; Jumper et al., 2021). It is worth mentioning that distinct from previous works that can only generate backbone atom positions (Jin et al., 2021; Kong et al., 2022; Luo et al., 2022), our model is capable of full atom position generation by attaching the corresponding sidechain to each residue. In specific, we can enable the decoder to generate four additional torsion angles ($\chi_1, \chi_2, \chi_3, \chi_4$) (McPartlon & Xu, 2022) that specify the geometry of sidechain atoms and reconstruct sidechain atom positions together with backbone atom positions.

**Training Objective.** We denote the reconstructed full backbone atom positions at $t^{\text{th}}$ layer as $\{\boldsymbol{x}_{ij}^t\}$, where $j \in \{1, 2, 3\}$ is the index of backbone atoms $(N, C_\alpha, C)$. We use the superscript $\text{true}$ to denote the ground truth value for simplicity. The whole network can be jointly optimized, by defining an invariant cross-entropy loss $\ell_{\text{ce}}$ over type distributions and an invariant frame align loss (Jumper et al., 2021) over full backbone atom positions at each translation layer:

$$\mathcal{L}_{\text{type}} = \frac{1}{T \cdot N} \sum_{t=1}^T \sum_{i=1}^N \ell_{\text{ce}}(\boldsymbol{s}_i^t, \boldsymbol{s}_i^{\text{true}}), \tag{10}$$

$$\mathcal{L}_{\text{pos}} = \frac{1}{T \cdot N \cdot 3N} \sum_{t=1}^T \sum_{k=1}^N \sum_{i,j} \|\rho_k^t(\boldsymbol{x}_{ij}^t) - \rho_k^{\text{true}}(\boldsymbol{x}_{ij}^{\text{true}})\|_2, \tag{11}$$

where $\rho_k^t(\boldsymbol{x}_{ij}^t) = (\boldsymbol{O}_k^t)^{-1}(\boldsymbol{x}_{ij}^t - \boldsymbol{x}_k^t)$ transforms the backbone coordinates from global coordinate system into local frame of $k^{\text{th}}$ residue frame at step $t$ (specified by $\boldsymbol{x}_k^t$ and $\boldsymbol{O}_k^t$), and so does $\rho_k^{\text{true}}$. The loss defined in Eq. 11 essentially calculates the discrepancy of all backbone positions between the prediction and the ground truth, *by aligning each residue frame one by one*. As all backbone coordinates are transformed into local frames, the loss will stay invariant when two protein structures differ by an arbitrary rotation and an arbitrary translation. Combining Eq. 10 and Eq. 11, the final training objective is $\mathcal{L} = \mathcal{L}_{\text{type}} + \mathcal{L}_{\text{pos}}$.

## 4 EXPERIMENTS

Following previous works (Jin et al., 2021; Jing et al., 2021; Anand & Achim, 2022), we conduct extensive experiments and evaluate the proposed PROTSEED on the following three tasks: **Antibody CDR Co-Design** (Section 4.1), **Protein Sequence-Structure Co-Design** (Section 4.2), and **Fixed Backbone Sequence Design** (Section 4.3). We also show cases where PROTSEED successfully conducts *de novo* protein sequence design with new folds in Section 4.4. We describe all experimental setups and results in task-specific sections.

### 4.1 ANTIBODY CDR CO-DESIGN

**Setup.** The first task is to design CDR sequence and structure of antibodies, where the context features are amino acid types as well as inter-residue distances derived from antibody-antigen complexes with CDRs removed. The initial protein structure $\mathcal{P}^0$ is set as the complex structure except for CDRs, which are randomly initialized. We retrieve antibody-antigen complexes in Apr. 2022 from Structural Antibody Database (SAbDab) (Dunbar et al., 2014), and remove incomplete or redundant complexes, resulting in a subset containing 2,900 complex structures. Following Jin et al. (2021), we focus on the design of heavy chain CDRs and curate three data splits for each type of CDRs (denoted as H1, H2, H3) by clustering corresponding CDR sequences via MMseqs2 (Steinegger & Söding, 2017) with 40% sequence identity. In total, there are 641, 963, and 1646 clusters for CDR H1, H2, and H3. The clusters are then divided into training, validation, and test set with a ratio of 8:1:1.

We compare PROTSEED with the following three baselines. RosettaAntibodyDesign (**RAbD**) (Adolf-Bryfogle et al., 2018) is a physics-based antibody design software. **GNN** is an autoregressive model that co-designs sequence and structure similar to Jin et al. (2021). We note that we can not directly compare PROTSEED with Jin et al. (2021) as the setting is different. **Diffusion** (Luo et al., 2022)

is a diffusion-based method that achieves state-of-the-art performance on antibody design recently. Following previous works (Jin et al., 2021; Luo et al., 2022), we use three metrics to evaluate the quality of designed CDRs: (1) Perplexity (**PPL**) measures the inverse likelihood of native sequences in the predicted sequence distribution, and a lower PPL stands for the higher likelihood. Note that PPL should be calculated *for each sequence first and then averaged over the test set.* For methods that do not define joint distributions over sequences strictly (e.g., DIFFUSION), we use the final sequence distribution to calculate the approximated PPL. (2) **RMSD** is the Root of Mean Squared Deviation of $C_\alpha$ between generated CDRs and ground-truth CDRs *with antibody frameworks aligned* (Ruffolo et al., 2022). A lower RMSD stands for a smaller discrepancy compared to native CDRs, which indicates a better CDR structure to bind the antigen. (3) Amino acid recovery rate (**AAR**) is the sequence identity between generated and ground-truth CDRs.

Table 1: PPL, RMSD and AAR of different approaches on the antibody CDR co-design task. (↑): the higher the better. (↓): the lower the better.

| CDR | PPL (↓) | | | RMSD (Å, ↓) | | | AAR (%, ↑) | | |
|---|---|---|---|---|---|---|---|---|---|
| | H1 | H2 | H3 | H1 | H2 | H3 | H1 | H2 | H3 |
| RABD | — | — | — | 3.06 (.33) | 2.95 (.23) | 5.58 (.32) | 27.36 (1.89) | 47.11 (1.74) | 22.91 (2.18) |
| GNN | 5.74 (.45) | 7.64 (.53) | 13.91 (.39) | 2.73 (.06) | 3.14 (.13) | 4.06 (.14) | 61.49 (.27) | 52.40 (.08) | 29.83 (.32) |
| DIFFUSION | 4.72 (.09) | 6.28 (.10) | 12.27 (.11) | 1.94 (.12) | 1.92 (.37) | 3.95 (.01) | 66.38 (.95) | 55.61 (.03) | 31.22 (.79) |
| **PROTSEED** | **4.43 (.40)** | **5.94 (.17)** | **10.88 (.38)** | **1.24 (.03)** | **1.11 (.04)** | **3.19 (.03)** | **70.22 (.98)** | **63.53 (.85)** | **39.27 (.77)** |

**Results.** We notice that scripts used to calcuate the above metrics are inconsistent in previous works. For a fair comparison, we implement all the baselines and run each model three times with different random seeds. Following previous works (Jin et al., 2021; Luo et al., 2022), the length of the CDR is set to be identical to the length of the ground-truth CDR for simplicity, and we sample 100 candidates with the lowest perplexity for each CDR for machine learning-based methods. We report the mean and standard deviation of the above metrics on the test set in Table 1. Numerical results indicate that PROTSEED consistently outperforms previous state-of-the-art baselines by a clear margin on all three metrics for each type of CDRs, which confirms PROTSEED's ability to co-design antibodies conditioned on existing binding structures. In particular, as an energy-based method that performs graft antibody design, the performance of RABD is inferior to data-driven approaches. DIFFUSION outperforms GNN as it is equivariant and models the orientation of residues similar to our model, but its performance still falls short of ours. It is worth mentioning that the performance of all models on CDR H3 is worse than that on the other two CDR types, as CDR H3 is the most diverse region in an antibody that is critical to antigen binding. We present generated samples on CDR-H3 sequence-structure co-design and their sidechain interactions with binding antigens in Figure 3(b).

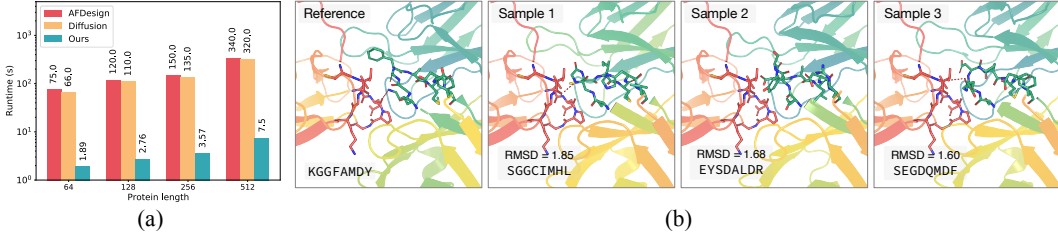

Figure 3: (a) Average runtime of three approaches on proteins of different sizes. Our PROTSEED runs orders of magnitude faster than the gradient-based method AFDESIGN and the diffusion-based method DIFFUSION. (b) Examples of CDR-H3 sequence and structure co-designed by our method. The PDB ID of the reference complex is 6FLA, and the antigen is Dengue Virus. The sidechain interactions between the antigen (red) and the antibody (green) are highlighted.

## 4.2 PROTEIN SEQUENCE AND STRUCTURE CO-DESIGN

**Setup.** This task evaluates models' capability to design protein sequence and structure conditioned on context features known *a priori*. The task is brought into attention by Anand & Achim (2022) recently, but there is no rigorous benchmark in the field of machine learning to the best of our knowledge. To this end, we collect 31,877 protein structures from the non-redundant set of CATH S40 (Orengo et al., 1997), calculate amino acid secondary structure by DSSP (Kabsch & Sander, 1983), and split the data set into training, validation, and test set at the topology level with a 90/5/5 ratio. We take secondary

structure annotations as single features and binary contact matrices as pair features.[1] This setting represents the situation where biologists already know the topology of desired proteins (Dou et al., 2018; Shen et al., 2018), and want the model to design novel proteins without specifying structures.

For this task, we compare PROTSEED with the following two baselines that we believe best describe the current landscape of this task. **AFDesign**[2] is a hallucination-based (Wang et al., 2021) method that generates protein sequence and structure by iteratively performing gradient descent in sequence space guided by AlphaFold2. We adopt AFDESIGN as one of our baselines because AlphaFold2 is the state-of-the-art protein structure prediction model, and it outperforms other hallucination-based methods. **Diffusion** (Anand & Achim, 2022) is another diffusion-based model that generates protein structure, sequence, and rotamers sequentially, and it is not restricted to antibodies. We use the **PPL**, **RMSD**, and **AAR** metrics introduced in Section 4.1 to evaluate the fidelity of designed proteins.

**Results.** Since AFDESIGN is incaple of handling discrete contact matrices, we set the loss function that guides the gradient descent as the distogram loss computed by a pre-trained AlphaFold2 (Jumper et al., 2021). For DIFFUSION, we implement it by ourselves as its source code is not released yet. We run each model for three times with different random seeds

Table 2: PPL, RMSD and AAR of different approaches on the protein co-design task. (↑): the higher the better. (↓): the lower the better.

| Method | PPL (↓) | RMSD (Å, ↓) | AAR (%, ↑) |
|---|---|---|---|
| AFDESIGN | — | 3.47 (.11) | 12.05 (.28) |
| DIFFUSION | 11.63 (.08) | 2.33 (.37) | 24.97 (.32) |
| PROTSEED | **8.87 (.17)** | **1.29 (.02)** | **33.10 (.34)** |

and report the mean and standard deviation of the above metrics on the test set. As shown in Table 2, PROTSEED outperforms all the baselines on all three metrics significantly, which demonstrates its capability to generate proteins of high fidelity as regards both sequence and structure, and it is not limited to specific domain topologies. The performance of AFDESIGN falls short of other methods as it relies on gradient descent to optimize the sequence, which is prone to getting stuck in the local optimum due to the rough landscape of the loss function. The performance of DIFFUSION is also inferior to ours, as it generates protein using three separate diffusion models and fails to cross-condition on structure and sequence. In contrast, PROTSEED updates sequence and structure interdependently in an end-to-end manner.

To demonstrate the efficiency of our method, we test inference stage of different approaches using a single V100 GPU card on the same machine, and present average runtime of these methods on proteins of different sizes. As indicated by Figure 3(a), our PROTSEED runs orders of magnitude faster than two baseline models on all four protein sizes, as both AFDESIGN and DIFFUSION rely on time-consuming Monte Carlo sampling steps.

Table 3: PPL and AAR of different approaches on the fixed backbone sequence design task. (↑): the higher the better. (↓): the lower the better. Results of baselines are taken from Jing et al. (2021).

| Method | PPL (↓) | | | AAR (%, ↑) | | |
|---|---|---|---|---|---|---|
| | Short | Single-chain | All | Short | Single-chain | All |
| GVP-TRANSFORMER | 8.94 | 8.67 | 6.70 | 27.3 | 28.3 | 36.5 |
| STRUCTURED GNN | 8.31 | 8.88 | 6.55 | 28.4 | 28.1 | 37.3 |
| GVP-GNN | **7.10** | 7.44 | **5.29** | 32.1 | 32.0 | 40.2 |
| PROTSEED | 7.32 | **7.38** | 5.60 | **34.8** | **34.1** | **43.8** |

## 4.3 FIXED BACKBONE SEQUENCE DESIGN

**Setup.** The third task is to design protein sequences that can fold into given backbone structures, which is known as fixed backbone design. In this task, context features are dihedral angles and inter-residue distances derived solely from backbone coordinates (Jing et al., 2021), and protein structures are fixed as ground truth in the decoder. We use the CATH 4.2 dataset curated by Ingraham et al. (2019), and follow all experimental settings of Jing et al. (2021) rigorously for a fair comparison, i.e., using the same data splits and evaluation settings according to their official implementation. We compare PROTSEED with three baselines. Specifically, **Structured GNN** is an improved version of Ingraham et al. (2019). **GVP-GNN** (Jing et al., 2021) and **GVP-Transformer** (Hsu et al., 2022)

---

[1]In this work, we say two residues are in contact if the distance between two $C_\alpha$ is within 8 Å.
[2]https://github.com/sokrypton/ColabDesign

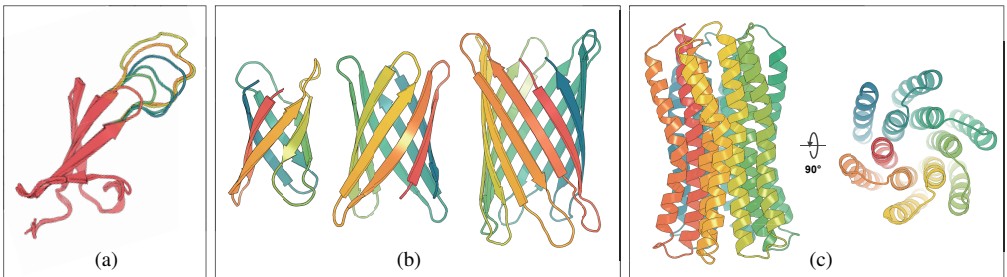

Figure 4: Example of novel proteins designed by PROTSEED. (a) Extending the loop of a native protein (marked in red). (b) Novel $\beta-$barrel design with different sizes. (c) Transmembrane protein complex design with a custom number of (twelve) $\alpha-$helices.

are state-of-the-art methods for fixed backbone design built upon Geometric Vector Perceptron (GVP) encoders (Jing et al., 2021). Note that since we can not afford training models on AlphaFold2-predicted data sets (Hsu et al., 2022) which requires hundreds of GPU days, we focus on the CATH data set for training. We evaluate the performance of all methods using **PPL** and **AAR** as introduced in Section 4.1, and drop the **RMSD** metric as structures are provided in this task.

**Results.** Following Jing et al. (2021), we report the evaluation results on three test splits including the full CATH 4.2 test set, the short subset (100 or fewer residues) and the single-chain subset. As shown in Table 3, PROTSEED achieves competitive results against the state-of-the-art method GVP-GNN in terms of the perplexity, and outperforms all baselines in terms of the amino acid recovery rate. The results indicate that PROTSEED is a quite general protein design framework and is also superior in designing protein sequences conditioned on desired backbone structures. Note that the state-of-the-art method GVP-TRANSFORMER is outperformed by GVP-GNN when trained solely on CATH data set, which is consistent with the results reported in the original paper (Hsu et al., 2022).

## 4.4 CASE STUDY

So far, we have evaluated PROTSEED's capability to design proteins of high fidelity on multiple settings. However, it remains unclear whether the proposed model can go beyond the topologies of existing proteins. To get insight into the proposed method and as a proof of concept, we manually construct a set of secondary structure annotations and contact features from scratch, and ask the model trained in the second task to perform *de novo* protein design based on the context features provided by us. In Figure 4, we show that our PROTSEED succeeds in altering loop lengths of existing proteins, designing novel proteins with idealized topologies, and designing novel protein complexes with a custom number of secondary structures. Notably, the designed structures are in close agreement with structures predicted by AlphaFold, taking the designed sequences as input. We further perform protein sequence and structure search against all available databases using FoldSeek (van Kempen et al., 2022) and BLAST (Altschul et al., 1990), and find these synthetic proteins are dissimilar to existing proteins regarding both sequence and structure. The case study serves as the first attempt to apply our model to *de novo* protein design in a more realistic setting, which reveals the possibility of PROTSEED being a powerful tool for protein design in biological research. We refer readers to Appendix C.1 for all details about the case study.

## 5 CONCLUSION AND FUTURE WORK

In this paper, we propose a novel principle for protein sequence and structure co-design called PROTSEED, which translates proteins in the joint sequence-structure space in an iterative and end-to-end manner. PROTSEED is capable of capturing the interplay of sequence, structure, and context features during the translation, and owns a much more efficient inference process thanks to the one-shot translation strategy. Extensive experiments over a wide range of protein design tasks show that PROTSEED outperforms previous state-of-the-art baselines by a large margin, confirming the superiority and generality of our method. Further case studies on *de novo* protein design demonstrate PROTSEED's potential for more practical applications in biological research. Future work includes extending PROTSEED to the scaffolding task (Trippe et al., 2022) and adopting latent variables (Kingma & Welling, 2013) to enable the context-free protein design.

REPRODUCIBILITY STATEMENT

For the sake of reproducibility, the pseudo-code of PROTSEED, the parameterization of SeqIPA, as well as hyper-parameters and implementation details are provided in Appendix B. All codes, datasets, and experimental environments will be released upon the acceptance of this work.

ACKNOWLEDGEMENT

We would like to thank all the reviewers for their insightful comments. Jian Tang is supported by Twitter, Intel, the Natural Sciences and Engineering Research Council (NSERC) Discovery Grant, the Canada CIFAR AI Chair Program, Samsung Electronics Co., Ltd., Amazon Faculty Research Award, Tencent AI Lab Rhino-Bird Gift Fund, an NRC Collaborative R&D Project (AI4D-CORE-06) as well as the IVADO Fundamental Research Project grant PRF-2019-3583139727.

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

## A EQUIVARIANCE PROPERTY OF SEQUENCE AND STRUCTURE TRANSLATION

We first quickly recap the process of sequence and structure translation in each translation layer. At $(t+1)^{\text{th}}$ layer, the decoder takes protein $\mathcal{P}^t = \{(\boldsymbol{s}_i^t, \boldsymbol{x}_i^t, \boldsymbol{O}_i^t)\}_{i=1}^N$ and context features $\{\boldsymbol{m}_i^t\}, \{\boldsymbol{z}_{ij}^t\}$ as the input. It encodes sequence-structure interplay and integrates all interactions into updated context features using SeqIPA adapted from Invariant Point Attention (IPA) (Jumper et al., 2021) in a way that its roto-translation invariant property is kept. The updates of $C_\alpha$ positions, frame orientations, and type distributions are then predicted based on updated context features. The whole process can be summarized as follows:

$$\boldsymbol{s}_i^{t+0.5} = \text{MLP}_e(\boldsymbol{s}_i^t), \tag{12}$$

$$\boldsymbol{m}_i^{t+1}, \boldsymbol{z}_{ij}^{t+1} = \text{SeqIPA}(\{\boldsymbol{m}_i^t\}, \{\boldsymbol{z}_{ij}^t\}, \{\boldsymbol{s}_i^{t+0.5}\}, \{\boldsymbol{x}_i^t\}, \{\boldsymbol{O}_i^t\}), \tag{13}$$

$$\hat{\boldsymbol{x}}_i^t = \text{MLP}_x(\boldsymbol{m}_i^{t+1}, \boldsymbol{m}_i^0), \quad \boldsymbol{x}_i^{t+1} = \boldsymbol{x}_i^t + \Delta\boldsymbol{x}_i^t = \boldsymbol{x}_i^t + \boldsymbol{O}_i^t \hat{\boldsymbol{x}}_i^t, \tag{14}$$

$$\hat{\boldsymbol{O}}_i^t = \text{convert}\left(\text{MLP}_o(\boldsymbol{m}_i^{t+1}, \boldsymbol{m}_i^0)\right), \quad \boldsymbol{O}_i^{t+1} = \boldsymbol{O}_i^t \hat{\boldsymbol{O}}_i^t, \tag{15}$$

$$\boldsymbol{s}_i^{t+1} = \text{softmax}\left(\lambda \cdot \text{MLP}_s(\boldsymbol{m}_i^{t+1}, \boldsymbol{m}_i^0, \boldsymbol{s}_i^{t+0.5})\right). \tag{16}$$

To derive the equivariance property of each translation step, we use three functions $\mathcal{X}, \mathcal{O}, \mathcal{S}$ to denote the network that predicts the $C_\alpha$ position translation, orientation translation, and sequence translation described above, respectively. Formally, we have:

$$\Delta\boldsymbol{x}_i^t = \mathcal{X}(\mathcal{P}^t), \tag{17}$$

$$\boldsymbol{O}_i^{t+1} = \mathcal{O}(\mathcal{P}^t), \tag{18}$$

$$\boldsymbol{s}_i^{t+1} = \mathcal{S}(\mathcal{P}^t). \tag{19}$$

Note that $\mathcal{X}, \mathcal{O}, \mathcal{S}$ also take $\{\boldsymbol{m}_i^t\}$ and $\{\boldsymbol{z}_{ij}^t\}$ as input and we omit these context features for simplicty, as they remain invariant to global rigid transformations. $\mathcal{X}, \mathcal{O}, \mathcal{S}$ are not separate networks, and they share the same input and the same SeqIPA, but are equipped with different MLPs. With the above definitions, we can derive the following proposition:

**Proposition 1** (Roto-Translation Equivariance). *Let $\mathcal{T}_{\boldsymbol{R},\boldsymbol{r}}$ denote any SE(3) transformation (rigid transformation) operating on the protein object $\mathcal{P}^t = \{(\boldsymbol{s}_i^t, \boldsymbol{x}_i^t, \boldsymbol{O}_i^t)\}_{i=1}^N$, with a rotation matrix $\boldsymbol{R} \in \text{SO}(3)$ and a translation vector $\boldsymbol{r} \in \mathbb{R}^3$. The function $\mathcal{X}, \mathcal{O}, \mathcal{S}$ satisfy the following equivariance properties:*

$$\mathcal{X} \circ \mathcal{T}_{\boldsymbol{R},\boldsymbol{r}}(\mathcal{P}^t) = \boldsymbol{R}\mathcal{X}(\mathcal{P}^t), \tag{20}$$

$$\mathcal{O} \circ \mathcal{T}_{\boldsymbol{R},\boldsymbol{r}}(\mathcal{P}^t) = \boldsymbol{R}\mathcal{O}(\mathcal{P}^t), \tag{21}$$

$$\mathcal{S} \circ \mathcal{T}_{\boldsymbol{R},\boldsymbol{r}}(\mathcal{P}^t) = \mathcal{S}(\mathcal{P}^t), \tag{22}$$

*where $\mathcal{T}_{\boldsymbol{R},\boldsymbol{r}}(\mathcal{P}^t) = \{(\boldsymbol{s}_i^t, \boldsymbol{x}_i^t + \boldsymbol{r}, \boldsymbol{R}\boldsymbol{O}_i^t)\}_{i=1}^N.$*

Intuitively, the proposition states that in each translation step, the updates of $C_\alpha$ positions and frame orientations are equivariant with respect to input protein structures, and the updates of type distributions are invariant.

*Proof.* We first prove that Eq. 20 holds. Notice that SeqIPA is aware of the orientations of the input structure, and the updated context features are invariant (Eq. 13). Therefore, the predicted deviation of $C_\alpha$ positions, i.e., $\hat{\boldsymbol{x}}_i^t$, is invariant (Eq. 14). Then, we have:

$$\mathcal{X} \circ \mathcal{T}_{\boldsymbol{R},\boldsymbol{r}}(\mathcal{P}^t) = \boldsymbol{R}\boldsymbol{O}_i^t \hat{\boldsymbol{x}}_i^t = \boldsymbol{R}\mathcal{X}(\mathcal{P}^t). \tag{23}$$

The Eq. 21 and Eq. 22 can be proved in a similar way. □

# B  MODEL DETAILS

## B.1  PSEUDO CODE

The pseudo code of PROTSEED is provided in Algorithm 1. The proposed PROTSEED consists of a trigonometry-aware encoder (Algorithm 1, line 2-7) that reasons geometrical constraints and interactions from context features, and a roto-translation equivariant decoder (Algorithm 1, line 10-19) that translates protein sequence and structure interdependently. Starting from the intial single features $\{m_i\} \in \mathbb{R}^{N \times c_m}$ and pair features $\{z_{ij}\} \in \mathbb{R}^{N \times N \times c_z}$, the whole model iteratively translates both protein sequence and structure into the desired state from random initialization (Algorithm 1, line 9). We note that the whole process does not require MCMC sampling, and runs much faster than autoregressive models and diffusion-based models.

The trigonometry-aware encoder is composed of a stack of $L$ encoding layers. Each layer takes single features $\{m_i^l\}$ and pair features $\{z_{ij}^l\}$ from the last layer as its input, and updates these features with novel attention mechanisms. After $L$ rounds of feature propagation, the updated single features and pair features serve as inputs to the decoder for joint protein sequence-structure design (Algorithm 1, line 8).

The roto-translation equivariant decoder iteratively refines the concrete 3D atom coordinates of the protein from random initialization, based on context features calculated by the encoder. The interplay of residue types, residue structures and context features during the decoding process is captured by a novel orientation-aware attention mechanism (SeqIPA, Appendix B.2). The equivariance property of the structure translation process is guaranteed by mapping invariant predictions in local frames to global frames with change of basis (Algorithm 1, line 13-16). It is worth mentioning that PROTSEED updates sequence and structure of all residues in an one-shot manner, leading to a much more efficient inference process (Algorithm 1, line 17).

---

**Algorithm 1** PROTSEED

---

**Require:** Initial single features $\{m_i\} \in \mathbb{R}^{N \times c_m}$ and pair features $\{z_{ij}\} \in \mathbb{R}^{N \times N \times c_z}$.
1: $m_i^0, z_{ij}^0 \leftarrow \mathrm{Linear}(m_i), \mathrm{Linear}(z_{ij})$                     ▷ $m_i^0 \in \mathbb{R}^c, z_{ij}^0 \in \mathbb{R}^c$
2: **for** $l \leftarrow 0$ to $L-1$ **do**
3:      $m_i^{l+1} \leftarrow \mathrm{MHA}(\{m_i^l\}, \{z_{ij}^l\})$                      ▷ Eq. 1
4:      $z_{ij}^{l+0.5} \leftarrow z_{ij}^l + \mathrm{Linear}(m_i^{l+1} \otimes m_j^{l+1})$             ▷ Eq. 2
5:      $z_{ij}^{l+0.75} \leftarrow z_{ij}^{l+0.5} + \mathrm{TriangleUpdate}_1(\{z_{ij}^{l+0.5}\})$         ▷ Eq. 4
6:      $z_{ij}^{l+1} \leftarrow z_{ij}^{l+0.75} + \mathrm{TriangleUpdate}_2(\{z_{ij}^{l+0.75}\})$         ▷ Eq. 5
7: **end for**
8: $m_i^0, z_{ij}^0 \leftarrow m_i^L, z_{ij}^L$                 ▷ Initialize context features for decoder
9: $\mathcal{P}^0 \leftarrow \{(s_i^0, x_i^0, O_i^0)\}_{i=1}^N \leftarrow \{(\frac{1}{20} \cdot \mathbf{1}, (0,0,0), I_3)\}_{i=1}^N$       ▷ Initialize protein $\mathcal{P}^0$
10: **for** $t \leftarrow 0$ to $T-1$ **do**
11:      $s_i^{t+0.5} \leftarrow \mathrm{MLP}_{\mathrm{e}}(s_i^t)$                     ▷ $s_i^{t+0.5} \in \mathbb{R}^c$
12:      $m_i^{t+1}, z_{ij}^{t+1} \leftarrow \mathrm{SeqIPA}(\{m_i^t\}, \{z_{ij}^t\}, \{s_i^{t+0.5}\}, \{x_i^t\}, \{O_i^t\})$    ▷ Eq. 6 and Section B.2
13:      $\hat{x}_i^t \leftarrow \mathrm{MLP}_{\mathrm{x}}(m_i^{t+1}, m_i^0)$         ▷ Deviation of $C_\alpha$ positions in local frame
14:      $x_i^{t+1} \leftarrow x_i^t + O_i^t \hat{x}_i^t$          ▷ Deviation of $C_\alpha$ positions in global frame
15:      $\hat{O}_i^t \leftarrow \mathrm{convert}\left(\mathrm{MLP}_{\mathrm{o}}(m_i^{t+1}, m_i^0)\right)$       ▷ Convert a quaternion to a rotation matrix
16:      $O_i^{t+1} \leftarrow O_i^t \hat{O}_i^t$
17:      $s_i^{t+1} \leftarrow \mathrm{softmax}\left(\lambda \cdot \mathrm{MLP}_{\mathrm{s}}(m_i^{t+1}, m_i^0, s_i^{t+0.5})\right)$        ▷ Eq. 9
18:      $\mathcal{P}^{t+1} \leftarrow \{(s_i^{t+1}, x_i^{t+1}, O_i^{t+1})\}_{i=1}^N$
19: **end for**
**Return:** The trajectory of the protein translation $\{\mathcal{P}^t\}_{t=1}^T$.

---

## B.2 PARAMETERIZATION OF SEQIPA

SeqIPA is adapted from the Invariant Point Attention (IPA) (Jumper et al., 2021), which takes residue types as the additional input to capture the interactions between current decoded sequences, structures, and the context features. We ensure that the additional input does not affect the invariance property of the IPA to make full use of its capacity. Specifically, we propose the following two strategies to parameterize the SeqIPA.

**SeqIPA-Addition.** Given that $\{s_i^{t+0.5}\}$ share the same dimensionality with $\{m_i^t\}$, a very simple strategy is to just add embeddings of residue types onto single representations. Following the original implementation of IPA, we leave the pair features unchanged in this approach.

$$m_i^{t+1}, z_{ij}^{t+1} = \text{SeqIPA}(\{m_i^t\}, \{z_{ij}^t\}, \{s_i^{t+0.5}\}, \{x_i^t\}, \{O_i^t\}) \tag{24}$$

$$= \text{IPA}(\{m_i^t + s_i^{t+0.5}\}, \{z_{ij}^t\}, \{x_i^t\}, \{O_i^t\}). \tag{25}$$

The above equations say that for SeqIPA-Addition, we just add the sequence embeddings $\{s_i^{t+0.5}\}$ onto the single representation $\{m_i^t\}$ (Eq.25), and feed four inputs to vanilla IPA (Jumper et al., 2021).

**SeqIPA-Attention.** Another more complicated strategy is to construct a new set of single representations and pair representations based on the embeddings of the current residue types. Then, we adopt a lightweight encoder similar to the encoder introduced in Section 3.2 to update $m_i^t$ and $z_{ij}^t$, which are then fed into the vanilla IPA module. We summarize the computation flow as follows:

$$\bar{m}_i, \bar{z}_{ij} = \text{Linear}(s_i^{t+0.5}), \text{Linear}(s_i^{t+0.5} + s_j^{t+0.5}) \tag{26}$$

$$\bar{m}_i, \bar{z}_{ij} = \text{Encoder}(\{\bar{m}_i\}, \{\bar{z}_{ij}\}), \tag{27}$$

$$m_i^{t+0.5}, z_{ij}^{t+0.5} = m_i^t + \bar{m}_i, z_{ij}^t + \bar{z}_{ij}, \tag{28}$$

$$m_i^{t+1}, z_{ij}^{t+1} = \text{SeqIPA}(\{m_i^t\}, \{z_{ij}^t\}, \{s_i^{t+0.5}\}, \{x_i^t\}, \{O_i^t\}) \tag{29}$$

$$= \text{IPA}(\{m_i^{t+0.5}\}, \{z_{ij}^{t+0.5}\}, \{x_i^t\}, \{O_i^t\}). \tag{30}$$

The above equations say that for SeqIPA-Attention, we leverage another lightweight encoder (Eq.27) similar to the encoder introduced in Section 3.2 to first update $m_i^t$ and $z_{ij}^t$ (Eq.28), and then feed updated inputs to vanilla IPA (Eq.30).

In practice, we find both strategies work well and their performance is on par with each other. To make the whole model lightweight, we adopt the first strategy across all the experiments in this work. We emphasize that the parameterization of the SeqIPA is quite flexible, as long as it can model interactions between sequences, structures, and context features, and is invariant to the global transformation of input structures. For the concrete computation flows of the IPA module, we refer readers to Algorithm 22 described in the supplementary material of Jumper et al. (2021).

## B.3 HYPER-PARAMETERS AND IMPLEMENTATION DETAILS

PROTSEED is implemented in Pytorch. The trigonometry-aware context encoder is implemented with $L = 8$ layers, and the sequence-structure decoder is implemented with $T = 8$ layers. The hidden dimension is set as 128 for pair features and 256 for single features across all modules.

For training, we use a learning rate of 0.001 with 2000 linear warmup iterations. We empirically find that proper learning rate warmup schedule can lead to faster convergence rate and higher performance. The model is optimized with Adam optimizer on four Tesla V100 GPU cards with distributed data parallel. The estimated time it takes to get a converged model is 24 hours.

For inference, the temperature of the sequence distribution, i.e., $\lambda$, controls the sharpness of the distribution. The larger $\lambda$ will lead to higher (better) AAR and higher (worse) PPL, and vice versa. Since it acts oppositely on AAR and PPL, we simply set it as 1 across the experiments.

All codes, datasets, and experimental environments will be released upon the acceptance of this work.

## B.4 FULL ATOM POSITIONS RECONSTRUCTION

The bond lengths and bond angles between backbone atoms are relatively conserved. The $C_\alpha$ is connected with N, C, and $C_\beta$ (except for Glycine which has a single hydrogen atom as its side

chain) atoms. The $C_\alpha$ forms a canonical orientation frame with respect to N, C, and $C_\beta$. Once we know the positions of the $C_\alpha$, and the orientation of the frame, the full backbone atom positions can be derived according to their averaged relative positions with respect to the $C_\alpha$ recorded in literature. The positions of all sidechain atoms of the 20 different amino acids can also be compactly specified by four torsion angles $(\chi_1, \chi_2, \chi_3, \chi_4)$ (Jumper et al., 2021; McPartlon & Xu, 2022), which also follow amino-acid specific distributions recorded in literature. All these recorded statistics can be found in https://git.scicore.unibas.ch/schwede/openstructure/-/raw/7102c63615b64735c4941278d92b554ec94415f8/modules/mol/alg/src/stereo_chemical_props.txt.

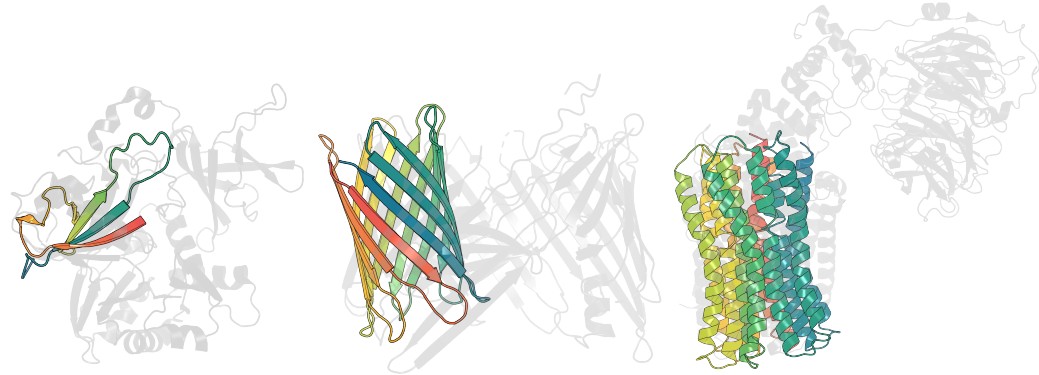

Figure 5: Superimposition of three generated proteins and their most similar proteins found in the PDB by FoldSeek. left: a novel protein with loop extended. middle: a novel $\beta-$barrel. right: a novel helical complex.

## C EXPERIMENTAL DETAILS

### C.1 CASE STUDY

We conduct three case studies to evaluate PROTSEED's capability to perform *de novo* protein design, including extending the loop of existing proteins, designing novel $\beta-$barrels, and designing novel helical complexes. Specifically, we manually curate a set of secondary structure annotations and contact features, and ask the model trained in the second task (Section 4.2) to generate novel proteins based on these context features. We elaborate the way we design context features for each setting.

**Extending the Loop.** In this setting, we start by calculating the secondary structure annotations and the contact matrix of an existing protein. We then insert $n$ consecutive "C" ("C" is the secondary structure annotation for the loop) letters into the original secondary structure annotations at the position where we want to extend the loop. Similarly, we insert $n$ consecutive rows and columns filled with zero into the original contact matrix. For a new-inserted residue indexed by $i$, we let it to be in contact with $i-2, i-1, i, i+1, i+2$.

**Designing Novel Beta Barrels.** In this setting, we grab a simple pattern of $\beta-$barrel proteins and then repeat this pattern multiple times to construct the contact features. The secondary structure annotations are also calculated by repeating the annotations of the pattern multiple times.

**Designing Novel Helical Complexes.** Similar to the second case, in this setting, we also take a simple pattern of helical complexes and construct the contact features by repeating it multiple times. The secondary structure annotations are all set to be "H".

In Figure 5, we show the superimposition of three novel proteins designed by PROTSEED against the most similar proteins in the PDB, one for each setting, which confirms the novelty of the designed proteins.

Table 4: Ablation study on the antibody CDR co-design task. (↑): the higher the better. (↓): the lower the better.

| CDR | PPL (↓) | | | RMSD (Å, ↓) | | | AAR (%, ↑) | | |
|---|---|---|---|---|---|---|---|---|---|
| | H1 | H2 | H3 | H1 | H2 | H3 | H1 | H2 | H3 |
| **PROTSEED** | **4.43** | **5.94** | **10.88** | **1.24** | **1.11** | **3.19** | **70.22** | **63.53** | **39.27** |
| No Sequence Interaction | 5.19 | 7.77 | 14.78 | 1.48 | 1.32 | 3.24 | 68.16 | 59.14 | 32.42 |
| Single Iteration Translation | 6.03 | 17.98 | 26.11 | 2.17 | 2.41 | 5.09 | 67.49 | 55.82 | 29.42 |
| No SeqIPA | 13.96 | 14.57 | 17.87 | 18.22 | 18.92 | 11.37 | 9.88 | 10.14 | 7.42 |

## C.2 ABLATION STUDY

To gain more insights into the effectiveness of each module in PROTSEED, we conduct additional ablation studies following the setting of the antibody CDR co-design in Section 4.1.

**Effectiveness of Cross-conditioning on Sequence and Structure.** In this ablation study (denoted as **No Sequence Interaction**), we replace the SeqIPA in Eq.6 with the vanilla IPA (Jumper et al., 2021), and directly predict the distribution of amino acid types for all residues at the last iteration of the decoding process using single features, i.e., $\{m_i^T\}$. The results shown in Table 4 indicate that when the model fails to cross-condition on both sequence and structure during the decoding, there is a significant performance drop in all three metrics, especially for PPL and AAR. This confirms the necessity to cross-condition on sequence and structure during the decoding, and the effectiveness of the proposed SeqIPA.

**Effectiveness of Iterative Translations.** In this ablation study (denoted as **Single Iteration Translation**), we replace the $T-$layer decoder with a single-layer decoder for protein translation. We note that the vanilla decoder of PROTSEED is composed of $T = 8$ consecutive translation layers with tied weights, and the number of trainable parameters of these two models are the same due to the weight tying. As indicated by Table 4, the single-layer decoder is outperformed by the vanilla decoder by a large margin on all metrics. Since the numbers of trainable parameters of these two models are the same, this is a fair comparison. The results justify the advantages of the iterative translation framework for protein sequence and structure co-design.

**Effectiveness of Context Feature Update.** In this ablation study (denoted as **No SeqIPA**), we freeze all context feature updates in the decoder and remove SeqIPA (Eq.6), and use the outputs of the encoder as the context features during the whole decoding process. As shown by Table 4, the performance degrades dramatically, which demonstrates that the context feature update plays a key role in PROTSEED.

# D    DISCUSSION

## D.1    CONTEXT FEATURES

In this work, we use the concept of context features (single features $\{m_i\}$ and pair features $\{z_{ij}\}$) as a formulation to unify the inputs of different protein design tasks. We note that $\{m_i\}$ and $\{z_{ij}\}$ vary from task to task, and for most well-defined tasks, they are easy to get. For example, in antibody CDR design tasks (Jin et al., 2021; Luo et al., 2022; Kong et al., 2022), they can be derived from antibody frameworks and structures of binding antigens. In the general protein design task proposed by Anand & Achim (2022), they can be derived from second structure annotations and contact maps provided by biologists. In scaffolding tasks, they can be derived from an starting motif (Trippe et al., 2022). The more context features (or constraints) the researchers specify, the more control they can have over the designed proteins. We refer readers to Dou et al. (2018); Shen et al. (2018) for two cases of protein design in real world scenarios.

## D.2    CONTEXT-FREE PROTEIN DESIGN

Researchers may be interested at generating novel proteins without relying on any context features, a.k.a. context-free protein design. We note that PROTSEED is a very general framework and can handle this situation with minor modifications. Specifically, we can adopt the Variational Autoencoder (VAE) framework (Kingma & Welling, 2013) and approximate the data distribution by learning to map proteins to latent vectors and reconstruct proteins from latents. In this scenario, the context features become the proteins sampled from target data distributions. Since this is out of the scope of this work, we leave it as our future work.

