# OpenReview forum: "Protein Sequence and Structure Co-Design with Equivariant Translation"
_ICLR.cc/2023/Conference — ICLR 2023 poster_

### Official Review · Reviewer_mVZ1 · 2022-10-24

**Confidence:** 4
**Correctness:** 4
**Technical Novelty And Significance:** 2
**Empirical Novelty And Significance:** 3
**Recommendation:** 6

**Clarity, Quality, Novelty And Reproducibility:**

The method description is quite clear. The proposed method is conceptually similar to previous co-design methods, with small but useful modifications in the generation step. The major contribution, perhaps, is demonstrating its utility over general proteins rather than antibodies.

**Strength And Weaknesses:**

Strength:
1. The proposed method improves upon the previous co-design method (Jin et al 2021) by generating all the amino acids in an iterative refinement process rather than left-to-right autoregressive process.
2. The inference cost is lower than diffusion models which requires thousands of generation steps.

Weakness:
1. For protein design tasks, the term "co-design" is somewhat misleading. The method does not generate protein structure (and sequence) from scratch. It is conditioned on the contact map + secondary structure, which is a fuzzier representation of the backbone structure but it contains many information of the structure. The contact map needs to be taken from an existing structure, although it can be specified manually for certain types of structures.

**Summary Of The Paper:**

This paper proposes an iterative method for protein sequence-structure co-design. In each generation step, the model first uses IPA network to update the 3D structure ($C_a$ coordinates and the orientation of its frame). It then predicts the amino acid types at all positions. The proposed method is applied to antibody CDR design (co-design) and protein design (fixed backbone and co-design) applications and showed state-of-the-art performance.

**Summary Of The Review:**

The paper is a good contribution to ML protein engineering field and I vote for acceptance. The methodology is similar to previous methods, but with useful modifications (e.g., iterative generation of amino acids rather than autoregressive). Moreover, it demonstrated its utility over general proteins (not just antibodies) and therefore has a larger impact.

---

> ### Author Response · Authors · 2022-11-09
> **Response to anonymous reviewer mVZ1**
>
> Thank you very much for your insightful comments and for affirming the contribution of our work. The responses to your concerns are attached below.
>
> ### 1. Concerns about the term "co-design"
> Thank you for raising these constructive points. **By using the word “co-design”, we aim to emphasize that our method can generate both protein sequences and structures, compared to some of the previous methods that only tackle one modality.** We use the concept of context features as a formulation to unify the inputs of different tasks under our framework.
> For the protein design task you mentioned, it indeed relies on given context features. The more context features (or constraints) the researchers specify, the more control they can have over the designed proteins. We refer reviewers and readers to [1,2] for protein design based on pre-defined constraints in real-world settings.
>
> We note that ProtSeed is a very general framework and can handle the context-free protein design with minor modifications. Basically, we can adopt the VAE framework [3] and approximate the data distribution by learning to map proteins to latent vectors and reconstruct proteins from latent vectors. In this scenario, the context features become the proteins sampled from target data distributions. This is out of the scope of this work and we leave it as our future work. **We have added a section in Appendix D discussing the concept of context features and the way to generate proteins without relying on context features in detail.**
>
> ### 2. Comments on differences with previous co-design method (Jin et al 2021 [4])
> In our humble view, in addition to the difference in the generation procedure (iterative translation in Protseed vs. autoregressive generation in [4]), our method is fundamentally and conceptually different with [4].
> We note that we use orientation frames (coordinates of c alpha atom and its orientation) to represent protein backbones, while [4] represents proteins as graphs.
> **We emphasize here that representing proteins as orientation frames is a more compact and efficient way, and it is super suitable for modeling the full atom positions of proteins(backbone atoms + sidechain atoms).** We note [4] can not model the full atom positions of proteins. **Please refer to Appendix B.4 for more details on full atom position reconstruction**.
>
> We hope the above responses address your concerns. Please let us know if you have other questions. We’re happy to further answer the questions.
>
> ### Reference
>
> [1] Dou, Jiayi, et al. "De novo design of a fluorescence-activating β-barrel." Nature 561.7724 (2018): 485-491.
>
> [2] Shen, Hao, et al. "De novo design of self-assembling helical protein filaments." Science 362.6415 (2018): 705-709.
>
> [3] Kingma, Diederik P., and Max Welling. "Auto-encoding variational bayes." arXiv preprint arXiv:1312.6114 (2013).
>
> [4] Jin, Wengong, et al. “Iterative refinement graph neural network for antibody sequence-structure co-design.”

---

> ### Author Response · Authors · 2022-11-29
> **Looking forward to more discussions**
>
> Dear reviewer,
>
> Since we have reached the middle stage of the AC-Reviewer-Author discussion phase and are towards the end of the overall discussion phase, we are thinking of sending this note since we have not heard back from you yet regarding our response to your concerns. We want to check if we are able to resolve all your concerns and if you have any further comments on our work.
> We are willing to address any additional concerns you might have, and we look forward to engaging in further discussions on our work.
>
> Sincerely,
>
> Authors of Paper 6221

---

### Official Review · Reviewer_UR24 · 2022-10-26

**Confidence:** 3
**Correctness:** 3
**Technical Novelty And Significance:** 2
**Empirical Novelty And Significance:** 3
**Recommendation:** 6

**Clarity, Quality, Novelty And Reproducibility:**

This paper is written clearly, and presents a novel model on interesting protein design tasks. Maybe future work can confirm whether the model can scale beyond the smaller dataset trained on.

**Strength And Weaknesses:**

Strengths:
- Paper uses well accepted metrics and implements recent results to make strong comparisons.
- SeqIPA is an interesting extension to the normal IPA module
- The results seem to be state of the art, beating out recent design models, at least on join sequence and structure design.

Weaknesses:
- The comparisons to fixed backbone design is weak, recent models like ESM-IF using larger datasets or prot-MPNN exists, and some effort should be made to compare to those.
- The work lack ablations, it's unclear which part of the model changes give such a large improvement.

**Summary Of The Paper:**

- Protseed introduces a model that can jointly model the sequence and structure information in order to do sequence design.
- Protseed tests against a suite of state of the art protein design benchmarks, like AlphaDesign, Diffusion, and GVP-GNN, on fixed backbone sequence design and antibody CDR design tasks.
- The model is able to get state of the art results and beat competitors on most tasks.

**Summary Of The Review:**

Because of the strong empirical results and good benchmarking, I would recommend an accept. Comparing to more recent models and on larger datasets would make this paper much stronger.

---

> ### Author Response · Authors · 2022-11-09
> **Response to anonymous reviewer UR24**
>
> Thank you for your comments and suggestions. We have conducted additional experiments to compare ProtSeed against the ProtMPNN on the fixed-backbone design tasks. We have also added ablation studies in the Appendix. The responses to your concerns are listed below.
>
> ### 1. Concerns about the fixed backbone experiments
>
> Thank you for pointing us to new baselines and new datasets. Those are very good suggestions.
>
> For the suggestion on larger datasets, as noted in Section 4.3 and in [1], the AlphaFold2-predicted data sets used by [1] require hundreds of GPU days for training. We are sorry that we can not afford to train ProtSeed on such a huge dataset currently, but we will definitely consider scaling ProtSeed on large datasets in the future via collaboration. We additionally note that the GVP-Transformer baseline used in our experiment is exactly the ESM-IF model, though it is trained on CATH 4.2.
>
> For the suggestion on comparing ProtSeed with the more recent model (ProtMPNN), we have conducted additional experiments according to your suggestions. We follow the setting of the inverse folding task (Section 4.3) and use the official implementation of ProteinMPNN [2] (https://github.com/dauparas/ProteinMPNN) with proper adaptions for bug fixing.
> We do not include any PSSM information (used in ProteinMPNN’s codebase) for a fair comparison. The results are presented in the following table. We will include these results in the final version.
>
> | Method   | PPL-short | PPL-single chain | PPL-All | AAR-short | AAR-single chain  | AAR-All |
> |----------|-----------|------------------|---------|-----------|-------------------|---------|
> | ProtSeed |    7.32   | **7.38**             | 5.60    | **34.8**      | **34.1**              | 43.8    |
> | GVP-GNN  | **7.10**      | 7.44             | **5.29**    | 32.1      | 32.0              | 40.2    |
> | ProtMPNN | 7.55      | 7.51             | 5.75    | 34.5      | 34.0              | **44.6**    |
>
> We notice that no single model beats other models on all six metrics, and our ProtSeed is competitive to GVP-GNN and ProteinMPNN, both of which are state-of-the-art inverse design models, confirming its effectiveness on the inverse folding task.
>
> ### 2. Concerns on the ablation studies
>
> We have conducted additional ablation studies according to your suggestions, and the results and discussions are available at Appendix C.2. In short, we studied:
>
> **(1)** the effectiveness of cross-conditioning on sequence and structure. The results show that **when the model fails to cross-condition on both sequence and structure during the decoding, there is a significant performance drop, which confirms the necessity to cross-condition on sequence and structure during the decoding.**
>
>  **(2)** the effectiveness of iterative translations against single-step translation. **The results justify the advantages of the iterative translation framework for protein sequence and structure co-design**.
>
> **(3)** the effectiveness of the context feature update during the decoding process. **The results demonstrate that the context feature update plays a key role in ProtSeed**.
>
> We hope the above responses address your concerns. Please let us know if you have other questions. We’re happy to further answer the questions.
>
> ### Reference
>
> [1] Hsu, Chloe, et al. "Learning inverse folding from millions of predicted structures." bioRxiv (2022).
>
> [2] Dauparas, Justas, et al. "Robust deep learning–based protein sequence design using ProteinMPNN." Science 378.6615 (2022): 49-56.

---

> ### Author Response · Authors · 2022-11-29
> **Looking forward to more discussions**
>
> Dear reviewer,
>
> Since we have reached the middle stage of the AC-Reviewer-Author discussion phase and are towards the end of the overall discussion phase, we are thinking of sending this note since we have not heard back from you yet regarding our response to your concerns. We want to check if we are able to resolve all your concerns and if you have any further comments on our work.
> We are willing to address any additional concerns you might have, and we look forward to engaging in further discussions on our work.
>
> Sincerely,
>
> Authors of Paper 6221

---

### Official Review · Reviewer_GDvK · 2022-11-01

**Confidence:** 5
**Clarity, Quality, Novelty And Reproducibility:** Please see above.
**Correctness:** 2
**Technical Novelty And Significance:** 2
**Empirical Novelty And Significance:** 2
**Recommendation:** 6

**Strength And Weaknesses:**

Strength:
1. The paper works on the protein sequence and structure co-design task, which is important and hot recently.
2. The method in this work is well-motivated and the modeling way is reasonable and solid.
3. The results show strong improvements over previous works on different tasks.

Weaknesses:
1. Most of the paper writing is good. However, in some claims and places, they are not appropriate enough. The authors mainly claim that the specific antibody prediction method, e.g., RefineGNN, is not general and is in a specific domain. However, in my view, their method is general, but they only show the application in antibodies. Besides, the sequence, structure, and rotamers sequentially generation are slow as the authors claimed, but the inconsistent and fanciful proteins may not be the truth. A good sequence can help structure, and a good structure can also help sequence, iteratively update sequence and structure is hard to say a negative point or "issue" as the authors mentioned.
2. From an overview of the technique, the trigonometry-aware encoder and the equivariant decoder are mostly taken from Alphafold2, though some modifications are taken. In terms of this, the novelty is hard to say as big as the authors claimed. Alphafold2 also worked on the protein (though they don't predict the sequences).
3. There are some places in the method questionable. (a). On the decoder side, within the encoder output, the decoder continuously updates the context feature $m$ and $z$, where the interactions of the context amino acids are involved in the encoding way. However, the update of the amino acids that need to predict seems to be not so clear and reasonable. For example, the $z$ feature is only performed between the amino acids in the context, but no consideration between the predicted amino acids (if I understand correctly of $z$). (b) The description of SeqIPA is not well presented. Though in supplementary (please append this as Appendix after the main text instead of supplementary) the authors try to explain, but the formulation is not shown. When compared with the original IPA, it is still not clear how to calculate with the five different inputs. (c) In eqn. 7 and 8, why the MLP takes m(t+1) and m(0) rather than updated m(t+1)? Is $\lambda$ necessary? Also more explanations may need for Algorithm 1. (d) In eqn.11, how $x_{ij}$ is calculated since in previous descriptions they are all based on the amino acid level instead of the atom level, this should be noted.
4. One disadvantage of this method is that they require the context of $m$ and $z$, amino acid and pair features. This somehow concerns me about the practice in real scenarios. The hard request of these two features may not be friendly for some tasks. For example, how to generate from a general distribution. The authors can provide more discussions.
5. Though the decoding speed is much improved. I would like to also know more about the training cost, since the method is much more complex than previous ones. For example, small encoders are also involved in the decoder side. Besides, as for the experimental comparisons, since the compared baselines are provided by the authors, the code should be published (also the implemented baseline code). Also, the ablation study is not provided, it is hard to see the contribution of each part.

**Summary Of The Paper:**

This paper proposes a framework for protein sequence and structure generation. The key idea is to involve the structure, sequence, and context together for encoding and decoding. The key modules are trigonometry-aware encoder and roto-translation equivariant decoder in a one-shot way. The experiments are conducted on three different tasks and the results show the ProtSeed is effective.

**Summary Of The Review:**

N/A

---

> ### Author Response · Authors · 2022-11-09
> **Response to anonymous reviewer GDvK (Part 1)**
>
> Thank you very much for your constructive and inspiring comments.
> We have conducted ablation studies, appended the Appendix after the main text, and revised the paper according to your suggestions. The detailed responses to your questions and concerns are listed below:
>
> ### 1. Concerns on potentially inappropriate or misleading claims on the previous protein design approaches.
> Thanks for raising this point. By mentioning the antibody-specific methods [1,2,3] and the sequential generation strategy of [4], we try to articulate the current landscape of the protein co-design task and the motivation behind the ProtSeed.
>
> For the **generalizability of antibody-specific methods**, I partially agree with you that these methods [1,2,3] can generalize to general proteins with proper adaptation. However, since these methods consist of modules tailored for antibodies, e.g., coarse-grained context (non-CDR) encoder [1], the adaptation can be non-trivial, and the effectiveness of these methods also remains unclear on general proteins. **Since “restricted to proteins with specific domain topologies” might be misleading, we have revised the sentence into “these approaches are tailored for antibodies and their effectiveness remains unclear on proteins with arbitrary domain topologies”.**
>
> For the **drawback of the sequential generation strategy[4]**, we are sorry for not describing it clearly and we believe there might be a misunderstanding.
> We note that the main model of [4] uses three separate diffusion models. It first generates structure, then generates sequence conditioned on structure, and finally generates rotamer conditioned on structure and sequence.
> In our humble view, **it can not generate structure conditioned on sequence, and can not iteratively update sequence and structure conditioned on each other** (a.k.a, cross-condition), **which is an issue discussed by the author in the original paper[4]**. To illustrate this point, we have added an ablation study in Appendix C.2. We remove the sequence and structure cross-conditioning, and generate sequences after the structure has been generated (quite like [4]). **We observe a clear performance drop with this modification.**
>
> By contrast, we iteratively update the sequence and structure in our model, trying to encourage the self-consistency of the proteins (the generated sequence folds to the generated backbone structure).
> **According to your suggestion, we delete the misleading word ‘fanciful’ and believe the word ‘inconsistent’ can best describe the situation.
> Thank you again for these suggestions.**
>
> ### 2. Concerns about the novelty of the work
> In our humble view, the main contribution of our work is orthogonal to that of AlphaFold2’s, though it is quite related to our work. In this work, we focus on tackling a new-rising problem, protein sequence and structure co-design, which is still under-explored in machine learning literature. To the best of our knowledge, this is the first machine-learning approach that is capable of cross-conditioning on sequence and structure for end-to-end protein sequence-structure co-design. We note that AlphaFold2 is incapable of this task.
>
> From a technical point of view, our trigonometry-aware encoder can serve as a general context encoder for protein modeling tasks, taking arbitrary single and pair features as input. It uses attention score weight tying to reduce the computational burden and does not require MSAs and templates as input.
> Our equivariant decoder is capable of cross-conditioning on sequence and structure and capturing interactions between them, both of which keep changing during the whole decoding process. We also propose a novel one-shot sequence and structure translation mechanism in contrast to the previous autoregressive or diffusion process, which largely accelerates the inference stage.

---

> > ### Author Response · Authors · 2022-11-09
> > **Response to anonymous reviewer GDvK (Part 2)**
> >
> > ### 3. Questions and Concerns on the method part
> > We are sorry for not making some method details clear due to the space limit. **We have revised the method and added discussions in the Appendix** to make it more clear according to your suggestions.
> >
> > **(a)** On the decoder side, within the encoder output, the decoder continuously updates the context feature m and z, where the interactions of the context amino acids are involved in the encoding way. However, the update of the amino acids that need to predict seems to be not so clear and reasonable.
> >
> > **Answer:** Sorry for not making this clear. In each decoding layer, we use SeqIPA to capture the interaction between pair feature ($z^t$), single feature ($m^t$), current structure $(x^t, O^t)$, and current sequence $(s^{t+0.5})$ (Eq. 6).
> > We assume all useful information is integrated into the updated single feature $(m^{t+1})$, which is used for protein translation (Eq. 789).
> > We don’t use pair features in Eq.789 as the dimension of z does not match with that of $x$, and $s$. Note that performing a cross-attention between $x$, $s$ and $z$ is a redundant operation as it has already been done in SeqIPA (Eq.6). Here, the updated pair features $z$ can be used to conduct pair-level tasks (e.g., distance prediction, contact prediction), which is out of the scope of this work.
> >
> > **(b)** Question on the details of SeqIPA
> >
> > **Answer:** Thank you for your suggestions. **We have appended the Appendix after the main text.** As described in Appendix B.2, we propose two strategies to parameterize the SeqIPA. For **SeqIPA-Addition**, we just add the sequence embedding $s_i^{t+0.5}$ onto the single representation (eq.25), and feed four inputs to vanilla IPA [6]. For **SeqIPA-Attention**, we leverage another lightweight encoder similar to the encoder introduced in Section 3.2 to update the $m$ and $z$ (eq.26-28), and feed updated features to vanilla IPA (eq.30). The computation flow of IPA follows the original verson [6].
> > **We have revised the Appendix B.2 to make the description of the SeqIPA more clear according to your suggestions.**
> >
> > **(c)** In eqn. 7 and 8, why the MLP takes m(t+1) and m(0) rather than updated m(t+1)? Is λ necessary? Also more explanations may need for Algorithm 1.
> >
> > **Answer:** We empirically find that taking $m_i^0$ as an additional input can lead to faster convergence, which is also a trick used in AF2’s implementation.  $\lambda$ is the temperature of the sequence distribution. The larger $\lambda$ will lead to higher (better) AAR and higher (worse) PPL, and vice versa. Since it acts oppositely on AAR and PPL, we simply set it as 1 across the experiments (see Appendxi B.3). **We have added more explanations on $\lambda$ (Appendix B.3) as well as on Algorithm 1 (Appendix B.1) according to your suggestions.**
> >
> > **(d)** In eqn.11, how x_ij is calculated since in previous descriptions they are all based on the amino acid level instead of the atom level, this should be noted.
> >
> > **Answer:** As noted in the main text (Section 3.3, right before the training objective), once we know the positions of C alpha, and orientation of the frame centered at C alpha, the full backbone atom positions can be derived according to their averaged relative positions with respect to C alpha recorded in literature [5,6] (these records can be found in https://git.scicore.unibas.ch/schwede/openstructure/-/raw/7102c63615b64735c4941278d92b554ec94415f8/modules/mol/alg/src/stereo_chemical_props.txt). **We have added a paragraph in Appendix B.4 to make it more clear for readers.**

---

> > > ### Author Response · Authors · 2022-11-09
> > > **Response to anonymous reviewer GDvK (Part 3)**
> > >
> > > ### 4. Concerns on the use of context features.
> > > Thanks for raising this great point! We first emphasize that the so-called context features (single feature m, pair feature z) are formulations we use to unify the input of different protein design tasks. We note that $m, z$ vary from task to task, and for most well-defined tasks, $m$ and $z$ are easy to get.
> > >
> > > For example, in antibody design[1,2,3], they can be derived from antibody framework and antigen structures. In protein design [4], they can be derived from second structure annotation and contact maps provided by biologists. In scaffolding, they can be derived from a starting motif [9]. The more context features (or constraints) the researchers specify, the more control they can have over the designed proteins.
> > > We refer readers to [7,8] for how biologists design proteins in real-world scenarios.
> > >
> > > The reviewer is interested in how we design proteins without any contexts or constraints (“i.e., how to generate from a general distribution”).
> > > We note that ProtSeed is a very general framework and can handle this situation with minor modifications.
> > > Basically, we can adopt the VAE framework [10] and approximate the data distribution by learning to map proteins to latent vectors and reconstruct proteins from latent vectors. In this scenario, the context features become the proteins sampled from target data distributions.
> > > Since this is out of the scope of this work, we leave it as our future work.
> > > **We have added a section in Appendix D discussing the concept of context features and the way to generate proteins from general distributions in detail.**
> > >
> > > ### 5. Concerns on the training cost, the reproducibility, and the ablation study.
> > > #### 5.1 **Training Cost**
> > >
> > > We note that similar to our model, both diffusion baseline models [3,4] use orientation-aware roto-translation invariant networks inspired by Alphafold, which use attention mechanism over single and pair features. (thus O(n^2 + n^3) where n is the number of residues)
> > > Therefore, the training costs of Protseed and [3,4] are roughly on the same level. For fair comparison, the GNN baseline adapt from RefineGNN also leverage the advanced attention mechanism over single and pair features, and runs slightly faster then ProtSeed and [3,4].
> > > In specific, the averaged per-protein (crop size =256) training speed of each model on a single V-100 card is as follows:
> > >
> > > | ProtSeed | Diffusion [3] | Diffusion [4] | GNN |
> > > |---|---|---|---|
> > > | 1.41s / per protein | 1.29s / per protein | 1.52s / per protein | 1.11 / per protein |
> > >
> > > For the first experiment (antibody design), we train all models for 6 hours. For the second and the third experiment, we train all models for 24 hours as the dataset is larger.
> > >
> > > #### 5.2 **Code / Reproducibility**
> > > We are currently refactoring the codebase into a more user-friendly version. **We have added a Reproducibility Statement paragraph before the reference. We promise all codes, datasets, and experimental environments (settings) will be released upon the acceptance of this work.**

---

> > > > ### Author Response · Authors · 2022-11-09
> > > > **Response to anonymous reviewer GDvK (Part 4)**
> > > >
> > > > #### 5.3 **Ablation Study**
> > > > We have conducted additional ablation studies according to your suggestions, and the results and discussions are available at Appendix C.2. In short, we studied:
> > > >
> > > > **(1)** the effectiveness of cross-conditioning on sequence and structure. The results show that **when the model fails to cross-condition on both sequence and structure during the decoding, there is a significant performance drop, which confirms the necessity to cross-condition on sequence and structure during the decoding.**
> > > >
> > > >  **(2)** the effectiveness of iterative translations against single-step translation. **The results justify the advantages of the iterative translation framework for protein sequence and structure co-design**.
> > > >
> > > > **(3)** the effectiveness of the context feature update during the decoding process. **The results demonstrate that the context feature update plays a key role in ProtSeed**.
> > > >
> > > > We really hope the above responses and revisions address your concerns. Please kindly let us know if you have any other questions. We’re always happy to further answer questions and improve the quality of the manuscript.
> > > >
> > > > ### Reference
> > > >
> > > > [1] Jin, Wengong, et al. “Iterative refinement graph neural network for antibody sequence-structure co-design.”
> > > >
> > > > [2] Kong, Xiangzhe, Wenbing Huang, and Yang Liu. “Conditional antibody design as 3d equivariant graph translation.”
> > > >
> > > > [3] Luo, Shitong, et al. "Antigen-specific antibody design and optimization with diffusion-based generative models.”
> > > >
> > > > [4] Anand, Namrata, and Tudor Achim. "Protein Structure and Sequence Generation with Equivariant Denoising Diffusion Probabilistic Models.”
> > > >
> > > > [5] McPartlon, Matthew, and Jinbo Xu. "AttnPacker: An end-to-end deep learning method for rotamer-free protein side-chain packing."
> > > >
> > > > [6] Jumper, John, et al. "Highly accurate protein structure prediction with AlphaFold." Nature 596.7873 (2021): 583-589.
> > > >
> > > > [7] Dou, Jiayi, et al. "De novo design of a fluorescence-activating β-barrel." Nature 561.7724 (2018): 485-491.
> > > >
> > > > [8] Shen, Hao, et al. "De novo design of self-assembling helical protein filaments." Science 362.6415 (2018): 705-709.
> > > >
> > > > [9] Trippe, Brian L., et al. "Diffusion probabilistic modeling of protein backbones in 3D for the motif-scaffolding problem." arXiv preprint arXiv:2206.04119 (2022).
> > > >
> > > > [10] Kingma, Diederik P., and Max Welling. "Auto-encoding variational bayes." arXiv preprint arXiv:1312.6114 (2013).

---

> > > > > ### Comment · Reviewer_GDvK · 2022-11-22
> > > > > **Thanks.**
> > > > >
> > > > > Thanks for the reset two parts.
> > > > > I look forward to more discussions.

---

> > > > > > ### Author Response · Authors · 2022-11-22
> > > > > > **Thanks.**
> > > > > >
> > > > > > Thanks for the kind reply. We are willing to further improve the quality of the manuscript.

---

> > > ### Comment · Reviewer_GDvK · 2022-11-22
> > > **Thanks**
> > >
> > > Thanks for this part. I feel it's much clearer now. Thanks for the authors for the kind rebuttal.
> > > One suggestion is still that a clear SeqIPA calculation (even an algorithm box) is required. The better way is not let the readers to go back find what is IPA calculated. If I didn't see wrong, there is no clear calculation equation of SeqIPA.

---

> > > > ### Author Response · Authors · 2022-11-22
> > > > **Response to the concerns about IPA**
> > > >
> > > > Dear Reviewer GDvK,
> > > >
> > > > Thank you again for your suggestion. We agree with you that we should not let readers find out what IPA is by themselves. **We will definitely include a clear SeqIPA calculation procedure in the final version**. (ICLR does not allow us to revise the paper currently.)

---

> > ### Comment · Reviewer_GDvK · 2022-11-22
> > **Thanks for the rebuttal**
> >
> > Dear Authors,
> >
> > I thank for your detailed and kind rebuttal. The description is clear and I am happy the first part is solved.
> > A clearer description is indeed important to not mislead readers and overclaim our contributions. I am happy that you have revised some. But you said "it can not generate structure conditioned on sequence, " is that some typos? or you want to say "it can not generate sequences conditioned on structures "?
> >
> > As for the novelty. I agree that you are working on similar method but mainly for protein design, and also alphafold2 didn't and can't do this. But from the technique novelty, this is not limited to what kind of applications, I guess we can keep both these views.
> > By the way, I am still concerned that the author want to claim the "first protein sequence-structure co-design". Even methods like Jin's antibody sequence-structure co-design can do similar things (I agree that you said "tailored"), but technically, I am not agreed that you said are hard to transfer or adaption. This is not clear to me.
> >
> > So we are on the same page that sequence and structure should be cross-conditioned for a better performance?

---

> > > ### Author Response · Authors · 2022-11-22
> > > **Response to reviewer GDvK's remaining concerns**
> > >
> > > Dear Reviewer GDvK,
> > >
> > > Thank you for your kind reply. By *"it can not generate structure conditioned on sequence"*, we mean that [1] fails to cross-condition on sequence and structure, as it generates protein in a "structure -> sequence -> rotamer" order. You see that when [1] generates the protein structure at the beginning, it does not leverage any sequence information as the sequence is **only generated after the structure is determined**. We will make it more clear in the paper.
> > >
> > > As for the novelty part, we sincerely thank the reviewer for helping us polish the manuscript. Yes, we are happy to keep both views in terms of Protseed's novelty.
> > > For the claim about the *"first protein sequence-structure co-design"*, we originally mean that we are the first to directly target this setting, but the claim might not be that accurate.
> > > We now **agree with the reviewer that Jin's method can be adapted to the co-design setting** (as we have done in the paper) with proper adaption (**although the efforts might be non-trivial**). We agree that *"first protein sequence-structure co-design"* might be a little bit over-claiming.
> > >
> > > We now really realize the importance of making these statements clear and avoiding potential misleading.
> > > **We are happy to adopt the reviewer's suggestion, and will revise these claims and remove inaccurate statements in the final version**. (ICLR does not allow us to revise them currently.)
> > >
> > > Yes, we now have achieved a consensus that cross-conditioning the sequence and structure can lead to better performance.

---

> ### Comment · Reviewer_GDvK · 2022-11-22
> **Improving in the final version**
>
> Dear Authors,
>
> Thanks for the quick response.
> I am satisfied with what you addressed now. As discussed here, please modify the claims and add calculations and other details in the final version. Increasing the score as we had good and nice communication. And also congratulations in advance. :)

---

> > ### Author Response · Authors · 2022-11-22
> > **Thanks!**
> >
> > We are very glad that our responses can address your concerns. Thank you again for your efforts in improving the quality of the work!

---

### Official Review · Reviewer_XDjj · 2022-11-04

**Confidence:** 4
**Correctness:** 3
**Technical Novelty And Significance:** 3
**Empirical Novelty And Significance:** 3
**Recommendation:** 6

**Clarity, Quality, Novelty And Reproducibility:**

The manuscript is well written, the method is reasonably novel, and the empirical performance is strong.

**Strength And Weaknesses:**

Strengths:
1. Good introduction and well-presented related work focusing on key aspects.
2. Good evaluation results compared to previous methods
3. A novel approach for solving the structure and sequence of protein and study for the de novo protein design

Weaknesses:
1. Lack of specific hyperparameter and training details
2. No mention that code and trained models will be released
3. The application of the antibody CDR design task is clear, but the relevance of the other two tasks is murkier. When would detailed structural specifications be available, but without an existing sequence that forms that fold?
4. The authors note a difference between their evaluation and the published scores for Diffusion, GNN and RABD models (Table 1). However, the concern remains that the Diffusion model shows significantly better RMSD scores compared to the scores in this paper. What is the source of this discrepancy?
5. ProtSeed improves over previous methods in terms of PPL, RMSD, and AAR. However, because the task is to design functional proteins, these metrics may not reflect true model performance, because a model with higher RMSD may, in fact, produce (more) functional proteins while also being worse at predicting structure. It would be helpful to provide some additional, functional-oriented, evaluation. For example, by comparing the protein-protein interaction patterns and charge distributions between the designs and the ground truth.
6. AAR is surprisingly low for all of these methods, including ProtSeed. If only 40% or less of amino acids are correctly recovered, how confident can we be that any of these methods actually produce good designs?
7. I’m not convinced by AF2 structure prediction as an evaluation. It isn’t surprising that AF2 would predict the designed proteins to fold as intended, because all of these models are trained on the same PDB data and, therefore, probably have the similar pathologies.

Other minor comments:
1. Figure 4. – To keep a uniform style among all figures, a, b c marks could be moved outside of the figures as in Fig. 1-3
2. Table 3 – highlight best results as in table 1 and 2


**Summary Of The Paper:**

The authors present a method for protein sequence and structure codesign. Current methods use either autoregressive or diffusion-based models, which the authors claim are computationally expensive and produce suboptimal solutions. Sequence generative models can produce good designs, but do not model protein structures. The authors address the above issue with their method ProtSeed. ProtSeed uses a triangular-aware encoder to learn geometrical constrain from context features and roto-translation equivariant decoder that iteratively improves protein structure followed by an MLP to decode the amino acid identities from the structure. The author performed experiments on the structural antibody database and 2 protein design benchmarks.

**Summary Of The Review:**

A good paper about an interesting method. Hyperparameters and training details need to be discussed. The code and models should be released to facilitate reproduction and future work. Some discrepancies with prior work should also be addressed.

What would improve my score: include hyperparameter and training details. Resolve discrepancy with prior work.

---

> ### Author Response · Authors · 2022-11-09
> **Response to anonymous reviewer XDjj (Part 1)**
>
> We thank the reviewer for the constructive comments. We have expanded the discussions on hyperparameters and training details in Appendix B.3 according to your helpful suggestions. The responses to your concerns, especially those on the discrepancy with prior work, are listed below.
>
> ### 1. Concerns on hyperparameters, training details, codes, and trained models (reproducibility).
> All essential hyperparameters and training details are elaborated in Appendix B.3 in the original manuscript. We are sorry for not making it clear to you. **We have further expanded the descriptions of hyperparameters and training details in Appendix B.3 according to your suggestions. We have appended the Appendix after the main text to make it more reachable.**
>
> For codes and reproducibility, we are currently refactoring the codebase into a more user-friendly version. **We have added a Reproducibility Statement paragraph before the reference. We promise all codes, datasets, and experimental environments (settings) will be released upon the acceptance of this work.**
> We really hope that ProtSeed can benefit the whole protein design community.
>
> ### 2. Concerns on the discrepancy of the performance of diffusion in Tabel 1.
> First, we emphasize the reasons that we can not directly take results from previous works are three folds. **(1)** We found the RMSD and PPL scores reported by different works are not calculated in a consistent way (elaborated in section 4.1). **(2)** We note that previous baselines [1,2,3] use different datasets processing schemes, which are not comparable directly. **(3)** We note that diffusion baseline [3] is a very recent model published on NeurIPS 2022, whose implementations haven’t been released by the submission deadline of ICLR 2023. **We thus implement it by ourselves to align with the results reported in the original paper as much as possible.**
>
> Second, we believe the discrepancy of the performance of diffusion in Table 1 is a result of several factors listed above. Especially, we note that **the reported score in the original paper[3] was obtained on a test set with only 20 antibodies**, which might be too small. We instead split the datasets with an 8:1:1 cluster ratio (clustered by sequence similarity), **which results in almost 200 test antibodies**. We believe such a processing scheme would lead to a more robust test set.
>
> Third, we found the official implementations of the diffusion baseline [3] were just released at the end of October, and we will consider integrating the official implementations as our baseline in the final version.
>
> ### 3. Concerns on definitions of three tasks and structural specifications.
> Thanks for raising this great point! We first emphasize that Protseed is a very general framework and is applicable to a wide range of protein design settings. We use the concept of context features as a formulation to unify the inputs of different tasks under our framework.
> Both the antibody CDR design task (the first) and the fixed-backbone design tasks (the third) are well-defined tasks in literature [1,2,3,5].
> The second task (protein design based on specified constraints) was recently brought into the attention by [4], which assumed context features are known a priori. We refer reviewers and readers to [6,7] for protein design based on pre-defined constraints in real-world settings.
>
> To generate proteins with novel folds without any existing sequence that forms that fold, the structural specifications can either be given according to domain knowledge [6,7] or by learning a context-free protein design model. **We have added a discussion in Appendix D to make it more clear.**

---

> > ### Author Response · Authors · 2022-11-09
> > **Response to anonymous reviewer XDjj (Part 2)**
> >
> > ### 4. Concerns about the evaluation metrics
> > Thanks for pointing us to new evaluation metrics. We follow the previous works published in machine learning literature [1,2,3] to evaluate the proposed method. Yes, you are right. The current evaluation metrics mainly measure how similar the designed proteins are to the ground truth, which is a persistent problem in literature [1,2,3]. We will actively survey more functional-aware metrics (e.g. binding free energy) and include them in our future version.
> >
> > For the AAR metrics, we empirically observe that a model with an AAR score between 30-40% can well recover most of the conserved sites in a protein, which are believed to be important to its stability.
> > Also, as you said in the fifth point, a single metric can not reflect the overall quality of designed proteins.
> > The validity of designed protein sequences and structures can further be measured with the RMSD scores against the reference proteins. We can also measure the RMSD scores between generated structures and AF2-predicted structures.
> >
> > ### 5. Concerns on the AF2 structure prediction as an evaluation.
> > This is a great point! I agree with you that using AF2 as a structure oracle is not the optimal solution. Therefore, we only use it as a proof of concept in the case study (Section 4.4) to show that the de novo designed proteins are reasonable to some extent (at least evaluated by the stoa structure prediction model). This is also a common practice for de novo protein design in biological literature [8, 9].
> > We shall resort to X-ray Crystallography or cryo-EM for the most rigorous evaluation. (of course, they are out of the scope of this work.)
> >
> > We hope the above responses address your concerns. Please let us know if you have other questions. We’re happy to further answer the questions.
> >
> > ### Reference
> > [1] Jin, Wengong, et al. “Iterative refinement graph neural network for antibody sequence-structure co-design.”
> >
> > [2] Kong, Xiangzhe, Wenbing Huang, and Yang Liu. “Conditional antibody design as 3d equivariant graph translation.”
> >
> > [3] Luo, Shitong, et al. "Antigen-specific antibody design and optimization with diffusion-based generative models.”
> >
> > [4] Anand, Namrata, and Tudor Achim. "Protein Structure and Sequence Generation with Equivariant Denoising Diffusion Probabilistic Models.”
> >
> > [5]Jing, Bowen, et al. "Learning from protein structure with geometric vector perceptrons." arXiv preprint arXiv:2009.01411 (2020).
> >
> > [6] Dou, Jiayi, et al. "De novo design of a fluorescence-activating β-barrel." Nature 561.7724 (2018): 485-491.
> >
> > [7] Shen, Hao, et al. "De novo design of self-assembling helical protein filaments." Science 362.6415 (2018): 705-709.
> >
> > [8] Wang, Jue, et al. "Deep learning methods for designing proteins scaffolding functional sites." bioRxiv (2021).
> >
> > [9] Dauparas, Justas, et al. "Robust deep learning–based protein sequence design using ProteinMPNN." Science 378.6615 (2022): 49-56.

---

> ### Author Response · Authors · 2022-11-29
> **Looking forward to more discussions**
>
> Dear reviewer,
>
> Since we have reached the middle stage of the AC-Reviewer-Author discussion phase and are towards the end of the overall discussion phase, we are thinking of sending this note since we have not heard back from you yet regarding our response to your concerns. We want to check if we are able to resolve all your concerns and if you have any further comments on our work.
> We are willing to address any additional concerns you might have, and we look forward to engaging in further discussions on our work.
>
> Sincerely,
>
> Authors of Paper 6221

---

### Author Response · Authors · 2022-11-09
**Response to all the reviewers, area chair, and readers**

We would like first to thank all the reviewers for your constructive and helpful reviews. We’ve revised the manuscript according to your suggestions. Specifically, we have made the following changes:
1. We **append the Appendix after the main text** to make it more reachable, and **change the color of the citation from pink to dark blue**.
2. We **add a Reproducibility Statement** paragraph before the reference. We promise all codes, datasets, and experimental environments will be released upon the acceptance of this work.
3. We **revise and expand the Appendix** to include more discussions about the work according to reviewers’ suggestions and also add ablation studies.
4. In the introduction section, we **revise the statement of antibody-specific methods** from “these approaches are restricted to proteins with specific domain topologies” to “these approaches are tailored for antibodies and their effectiveness remains unclear on proteins with arbitrary domain topologies”.

---

### Public Comment · ~Chentong_Wang1 · 2022-11-14
**some confusion about your cdr design benchmark**

I am very confused by your cdr benchmark, because your paper said that you do all rmsd metric by align designded structure on frame, but in TERATIVE REFINEMENT GRAPH NEURAL NETWORK FOR ANTIBODY SEQUENCE-STRUCTURE CO-DESIGN, Jin et al. (2021), they only use frame sequence as coarsed grained node, and the predict result is only the cdr coords and sequence, which means there is no possibility to test its performance on framed aligned rmsd since there is no way to know how predicted cdr graft to native frame, it would be preferred to describe a little how you recover backbone from coarse grainde node to do the frame align since without doing big modification, it cant be made by origin model

---

> ### Author Response · Authors · 2022-11-14
> **Response to the concern about the RefineGNN baseline**
>
> Hello Chentong,
>
> Thank you for mentioning the RefineGNN[1] baseline.
> You are right! The original RefineGNN adopts a coarse-grain representation for antibody frameworks, and can only predict the positions of CDR residues with a residue-level resolution. Therefore, the vanilla RefineGNN calculates RMSDs of predicted CDRs by directly superimposing them against the ground truth CDRs. The resulting RMSDs are not comparable to more recent baselines [2] directly.
>
> Recent deep learning-based methods [2] can model full-residue positions of antibodies, and consensus has been achieved that calculating RMSDs via superimposing the frameworks of antibodies is a more reasonable way. Also, [1] can not take binding antigens into consideration. Combining all these facts, the GNN baseline used in our experiment is adapted from [1] and is similar to [1], but without coarse-grain representation and is able to aggregate information from antigens. We note that [2] also uses this strategy to compare their model against [1].
> We will release the codes upon the acceptance of this work.
>
> Thank you again for your comments. Please let us know if you have other questions. We are happy to answer.
>
> ### Reference
>
> [1] Jin, Wengong, et al. “Iterative refinement graph neural network for antibody sequence-structure co-design.”
>
> [2] Luo, Shitong, et al. "Antigen-specific antibody design and optimization with diffusion-based generative models.”

---

> > ### Public Comment · ~Chentong_Wang1 · 2022-11-15
> > **Response to the author**
> >
> > Thank you for your reply, it make sense you reimplement jin's model, i guess you should put a piece of word in appendix or someplace about this thing, because it's very easy to be confused you use the original model as baseline to compare with yours, but actually you add residel-level graph to construct backbone and antigen too rather than origin. Another thing to be mentioned, i don't see there is comparison in [2] Luo's paper with [1] jin's paper? I guess they do not reimplement [1] to do a frame-aligned comparision, both version in biorxiv and openreview.
> >
> > Reference
> >
> > [1] Jin, Wengong, et al. “Iterative refinement graph neural network for antibody sequence-structure co-design.”
> >
> > [2] Luo, Shitong, et al. "Antigen-specific antibody design and optimization with diffusion-based generative models.”

---

> > > ### Author Response · Authors · 2022-11-15
> > > **Response to the concern about the RefineGNN baseline**
> > >
> > > Hi Chentong,
> > >
> > > Thank you for your reply. As noted in Section 4.1, we say that the GNN baseline we use is adapted from [1], because the **original model from [1] is not comparable with ours, and the setting is quite different**. **We have expanded the original sentence to make it more clear and describe the reasons for this.**
> > >
> > > For your second question (''i don't see there is comparison in [2] Luo's paper with [1] jin's paper''), we note that both our model and [2] are not directly comparable with [1] due to the reasons I mentioned above.
> > > **Both [2] and our work adopt the same strategy: developing a GNN baseline that co-designs sequences and structures in an alternating way, similar to [1].**
> > >
> > > I'm not sure how [2] implements the GNN model. But **computing RMSDs with frame-align superimposition is a necessary condition for a fair comparison**. Hope the above replies address you questions.
> > >
> > >
> > > ### Reference
> > >
> > > [1] Jin, Wengong, et al. “Iterative refinement graph neural network for antibody sequence-structure co-design.”
> > >
> > > [2] Luo, Shitong, et al. "Antigen-specific antibody design and optimization with diffusion-based generative models.”

---

### Decision · Program_Chairs · 2023-01-20

**Decision:**

Accept: poster

**Justification For Why Not Higher Score:**

Please see comments above from AC-reviewer meeting.

**Justification For Why Not Lower Score:**

The authors were able to improve this work during discussion per reviewers' comments.

**Metareview: Summary, Strengths And Weaknesses:**

In this work, the authors proposed a method, ProtSeed, that iteratively perform protein sequence and structure design guided by desirable coarse-grain features, through integrating IPA approach and predicting amino acids at positions across the structure in a one-shot fashion within each step to achieve an efficient overall process. The authors demonstrate their approach to have good performance in antibody design and related protein design tasks. Their method could improve over state-of-the-art in the field, and brings new ideas. Nonetheless, some of their description is not clear and more extensive discussion is needed. The authors were able to revise their paper during discussion following the reviewers' suggestions which helped to improve the quality of this work.

**Note From Pc:**

if the above contains the word "oral" or "spotlight" please see: "oral" presentation means -> notable-top-5% and "spotlight" means -> notable-top-25%. As stated in our emails, we are disassociating presentation type from AC recommendations

**Summary Of Ac-Reviewer Meeting:**

During the AC-reviewer meeting, the overall tone from reviewers on this work is positive. The reviewers agree that technical novelty of this work is sufficiently interesting that would merit a favorable view amongst the borderline submissions. Thus, at the end of the meeting the agreement is to recommend acceptance. Nonetheless, specific points that should be addressed by the authors (in a final version) are:

- co-design is misleading should be clarified, this is an important point shared by all reviewers and AC
- related to the previous point, this method that is conditioned on contact map is not as exciting as it seemed initially so please clarify
- how to evaluate these models is an issue for the field that the authors are encouraged to further discuss
- while combine existing method is not as novel, the techniques used are decently interesting
- this work's conceptual novelty is limited, so the authors are encouraged to further make contributions via innovate in this area

The AC would like to note that, AI for science is growing field where this paper is a more positive amongst other submissions, but while this work could pass the bar in comparison to peer submissions, the general feeling is that the authors should make sure to clarify the points as listed above, as integrating these changes are important for acceptance and should be carefully made in the final version.